# Automated next-generation profiling of genomic alterations in human cancers

Laurel A. Keefer[1], James R. White [1], Derrick E. Wood [1], Kelly M. R. Gerding[1], Kenneth C. Valkenburg[1], David Riley[1], Christopher Gault[1], Eniko Papp[1], Christine M. Vollmer[1], Amy Greer[1], James Hernandez[1], Paul M. McGregor III[1], Adriana Zingone [2], Bríd M. Ryan [2], Kristen Deak [3], Shannon J. McCall [3], Michael B. Datto[3], James L. Prescott[4], John F. Thompson [1], Gustavo C. Cerqueira[1], Siân Jones[1], John K. Simmons[1], Abigail McElhinny[1], Jennifer Dickey[1], Samuel V. Angiuoli[1], Luis A. Diaz Jr.[5], Victor E. Velculescu[6] & Mark Sausen [1]✉

The lack of validated, distributed comprehensive genomic profiling assays for patients with cancer inhibits access to precision oncology treatment. To address this, we describe elio tissue complete, which has been FDA-cleared for examination of 505 cancer-related genes. Independent analyses of clinically and biologically relevant sequence changes across 170 clinical tumor samples using MSK-IMPACT, FoundationOne, and PCR-based methods reveals a positive percent agreement of >97%. We observe high concordance with whole-exome sequencing for evaluation of tumor mutational burden for 307 solid tumors (Pearson r = 0.95) and comparison of the elio tissue complete microsatellite instability detection approach with an independent PCR assay for 223 samples displays a positive percent agreement of 99%. Finally, evaluation of amplifications and translocations against DNA- and RNA-based approaches exhibits >98% negative percent agreement and positive percent agreement of 86% and 82%, respectively. These methods provide an approach for pan-solid tumor comprehensive genomic profiling with high analytical performance.

[1] Personal Genome Diagnostics Inc., Baltimore, MD 21224, USA. [2] Laboratory of Human Carcinogenesis, Center for Cancer Research, National Cancer Institute, Bethesda, MD 20850, USA. [3] Department of Pathology, Duke University School of Medicine, Durham, NC 27710, USA. [4] PathGroup, Nashville, TN 37217, USA. [5] Department of Medicine, Memorial Sloan Kettering Cancer Center, New York, NY 10065, USA. [6] Sidney Kimmel Comprehensive Cancer Center, Johns Hopkins University School of Medicine, Baltimore, MD 21287, USA. ✉email: msausen@pgdx.com

High-complexity, comprehensive next-generation sequence analyses are changing the diagnostic landscape of oncology[1–5]. Multiple targeted therapies against proteins affected by genetic alterations have been shown to be safer and more effective than traditional chemotherapies when used in an appropriate patient population[6]. This has been successfully demonstrated for a number of therapeutics targeting the protein products of specific genes that are altered in human cancer, including the use of imatinib in chronic myeloid leukemias carrying the BCR-ABL fusion, trastuzumab in *ERBB2* (HER2/neu) amplified breast cancer, and vemurafenib in *BRAF*-mutated melanoma. Molecular alterations have also been shown to have a predictive or prognostic effect, such as the poor response to anti-*EGFR* (epidermal growth factor receptor) monoclonal antibodies in patients with mutations at codons 12 and 13 of *KRAS* in colorectal cancer[7]. Finally, the recently established connection between microsatellite instability (MSI) or high tumor mutation burden (TMB) across multiple solid tumor indications and durable patient response to immune checkpoint inhibitor therapies[8–12] necessitates additional testing for these genomic signature biomarkers. Because the mutations and mutational processes driving each tumor may be unique, identifying the genetic landscape of each patient's cancer is critical for the development of a personalized treatment plan that takes advantage of the growing number of targeted and immune therapies.

Historically, the approach to testing for the presence of targetable mutations or genomic signatures has required multiple single-analyte immunohistochemistry or polymerase chain reaction (PCR) assays which often only test for one alteration at a time. As guidelines recommend testing for more than ten alterations for non-small cell lung cancer patients[13], the available tumor tissues are often exhausted before actionable targets are identified[14]. Without fully testing potentially actionable alterations, some patients that could have been candidates for specific therapies would be left without these treatment options. This barrier to broad diagnostic analyses can be overcome with the use of comprehensive genomic profiling, where multiple actionable sequence mutations, structural variations, and genomic signatures are evaluated at once from a single tissue specimen. By removing the need for multiple tests, more patients may be assessed for known alterations linked to a targeted or immune therapy, thus expanding patient access to ever-growing numbers of precision medicine therapies.

While comprehensive genomic profiling has been available for some time in a few specialized laboratories[2,15,16], it has not yet achieved widespread clinical adoption[17] due to a lack of validated assays that could be scaled in local laboratories, together with inadequate reimbursement related to regulatory clearance[18]. Even though some labs may be capable of developing an NGS-based targeted assay, comprehensive genomic profiling analyses are difficult to develop due to the sophisticated bioinformatics analysis and interpretation. Integrating the laboratory methods with automated bioinformatics into a single multi-analyte test would allow patient samples and testing data to stay within the local laboratory ecosystem and expedite local laboratory use, thus improving widespread clinical adoption. In addition, offering a standardized analysis solution that has received regulatory clearance with a clear reimbursement strategy[19] across labs would ease interoperability and application of patient test results, positively impacting the ability to standardize treatment plans from comprehensive genomic profiling test results.

In this work, we describe the development and analytical validation of the PGDx elio tissue complete test, which comprises a 2.2 Mb targeted gene panel, a kitted sample preparation system, and an accompanying automated bioinformatics analysis platform to enable comprehensive genomic profiling of sequence and structural variants, as well as genomic signatures such as TMB and MSI in patients with solid tumors (Fig. 1).

## Results

**Overall approach.** We first sought to identify the optimal size of a targeted gene panel for accurate determination of TMB as compared to whole-exome sequencing (WES) analyses of cancer genomes. Although sequence alterations and MSI could be determined using small panels of several hundred kilobases (kb), accurate analyses of TMB require substantially larger genomic regions to accurately represent the exome-wide TMB levels[20]. To achieve this, we performed in silico analyses of WES data from The Cancer Genome Atlas (TCGA) project to evaluate the predictive capacity for TMB of targeted panels of varying sizes by random selection of coding exons ranging from 100 kb up to 2.5 megabases (Mb), in 100 kb intervals with 100 models per targeted interval evaluated[21]. Nonsynonymous mutations, including single base substitutions, insertions, deletions, and splice-site alterations with mutant allele fractions (MAFs) ≥10% were analyzed from 4174 cancers that have been approved for therapy or are in clinical trials with checkpoint inhibitor therapy (CPI), including lung ($n = 933$), colorectal ($n = 287$), melanoma ($n = 455$), bladder ($n = 392$), uterine/endometrial ($n = 444$), head and neck ($n = 472$), liver ($n = 354$), gastric ($n = 390$), and all other cancers ($n = 447$). Somatic mutations observed within each simulated targeted region were enumerated across the targeted panels of varying size. We defined the reference TMB as the number of alterations in a cancer exome divided by the exome length in Mb. To better estimate the performance of these targeted panels to estimate TMB in a clinical setting, we calculated the performance for the general metastatic cancer population through reweighting the tumor type-specific results based on the relative estimated number of late-stage cancer cases diagnosed each year[22]. These analyses suggested that to achieve an accurate estimation of TMB using a targeted panel compared to WES across the pan-solid tumor cohort (Pearson correlation >0.90 and Spearman rho >0.80), one would need a targeted panel at least 1.0 Mb in size (Supplementary Fig. 1). When evaluating performance across individual tumor types for a targeted panel of this size, the Pearson correlation ranged from 0.90 to 1.00 and the Spearman rho ranged from 0.63 to 0.96, likely due to the variability and composition of the overall sequence mutation burden across cancer types (Supplementary Fig. 1). Based on these findings, we designed a targeted panel comprising 1.3 Mb of coding regions across 505 genes (Supplementary Data 1) to enable accurate TMB analyses. In order to additionally assess structural variants and MSI in repeat regions across the genome, we also included an additional 0.9 Mb of intronic regions, especially those containing mononucleotide repeats, resulting in a combined targeted panel size of 2.2 Mb. Taken together, elio tissue complete enables comprehensive genomic profiling of sequence alterations, structural variants, and genomic signatures, yielding the highest clinical diagnostic potential of all FDA-cleared or approved multigene, decentralized solid tumor oncology diagnostic platforms (Supplementary Data 2).

**Training and validation of machine-learning-based identification of somatic sequence variants.** Identification of somatic sequence variants is the foundation for estimation of TMB as well as for detecting clinically actionable tumor alterations. Therefore, we sought to adapt our previously reported machine-learning-based method for somatic variant identification, PGDx Cerebro[23], in coding, intronic, and regulatory regions of the targeted panel. This approach examines >50 independent sequencing and mutation-specific features to develop a random forest classifier

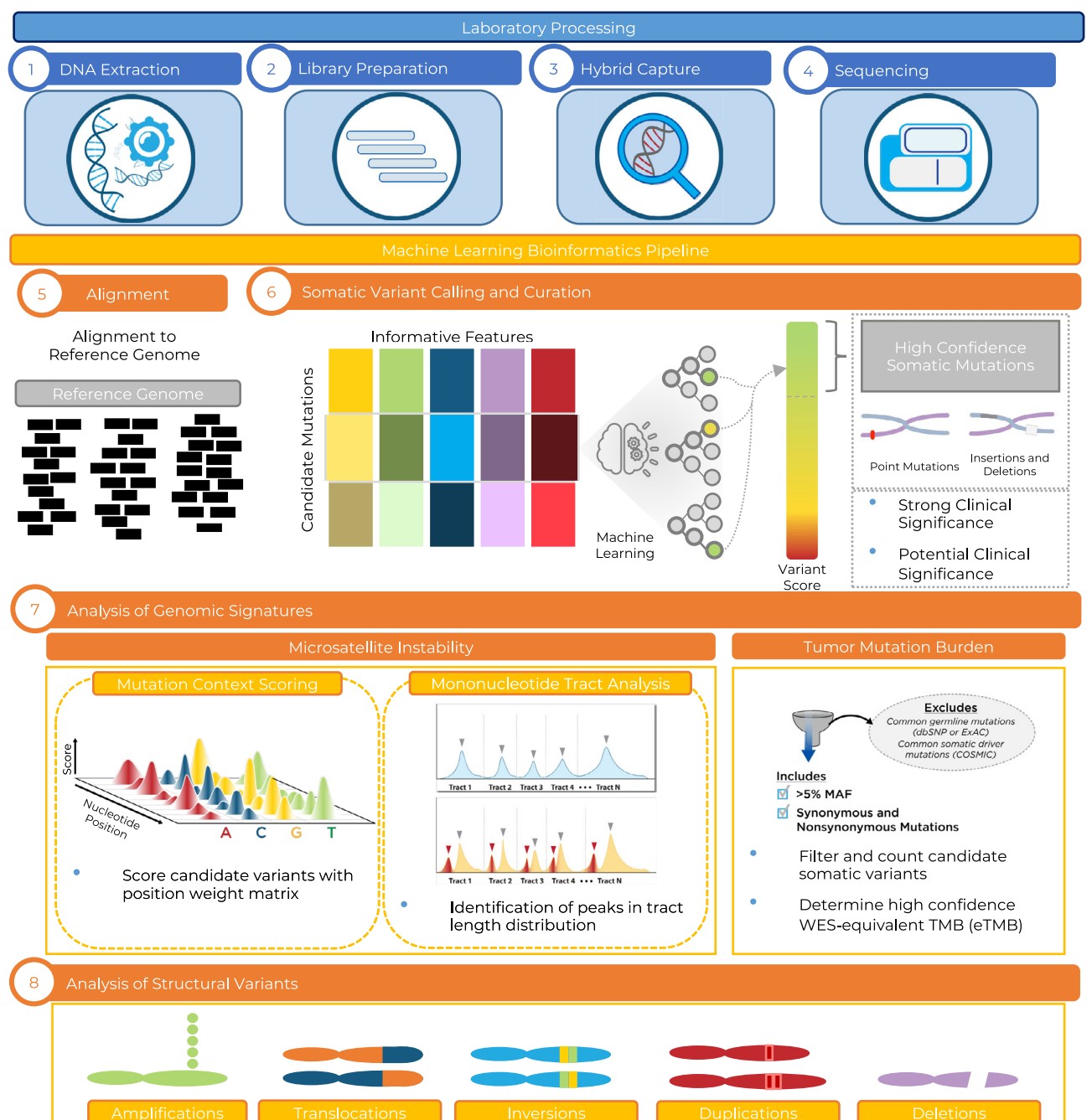

**Fig. 1 Overview of sample preparation and analysis for determination of sequence and structural alterations, TMB, and MSI by elio tissue complete.**
Genomic libraries were prepared using DNA extracted from cell line or FFPE tissue, and following hybrid capture and PCR amplification, DNA libraries were sequenced using the Illumina NextSeq. Next-generation sequencing data were analyzed using the VariantDx bioinformatics pipeline through alignment to the human reference genome assembly for the identification of sequence mutations, including single base substitutions (SBSs) and insertions/deletions (indels). Candidate variants were filtered through the PGDx Cerebro algorithm, designed to distinguish genuine somatic mutation calls from technical artifacts[23] and used to determine an elio-predicted exome tumor mutation burden (eTMB) score. Microsatellite status was determined using 68 mononucleotide tracts and specific sequence mutation contexts. Structural variants were identified with the Digital Karyotyping (DK)[29] and Personalized Analysis of Rearranged Ends (PARE)[29] algorithms. TMB tumor mutation burden, MSI microsatellite instability, MAF mutant allele fraction, WES whole-exome sequencing.

that can score a variant as somatic. To adapt PGDx Cerebro for targeted, tumor-only NGS analyses, we created a training set for this classifier by spiking a total of >124,000 in silico variants into a dataset of tumor-adjacent noncancerous DNA samples that were formalin-fixed, and paraffin-embedded (FFPE). The random forest classifier was applied to these training data, resulting in a

model optimized for somatic variant identification using elio tissue complete.

To validate this variant calling model, we analyzed 112 FFPE tumor samples using elio tissue complete as well as one of two targeted NGS assays (MSK-IMPACT, $n = 42$ or FoundationOne, $n = 70$[24]) (Fig. 2a–c, Supplementary Fig. 2, and Supplementary

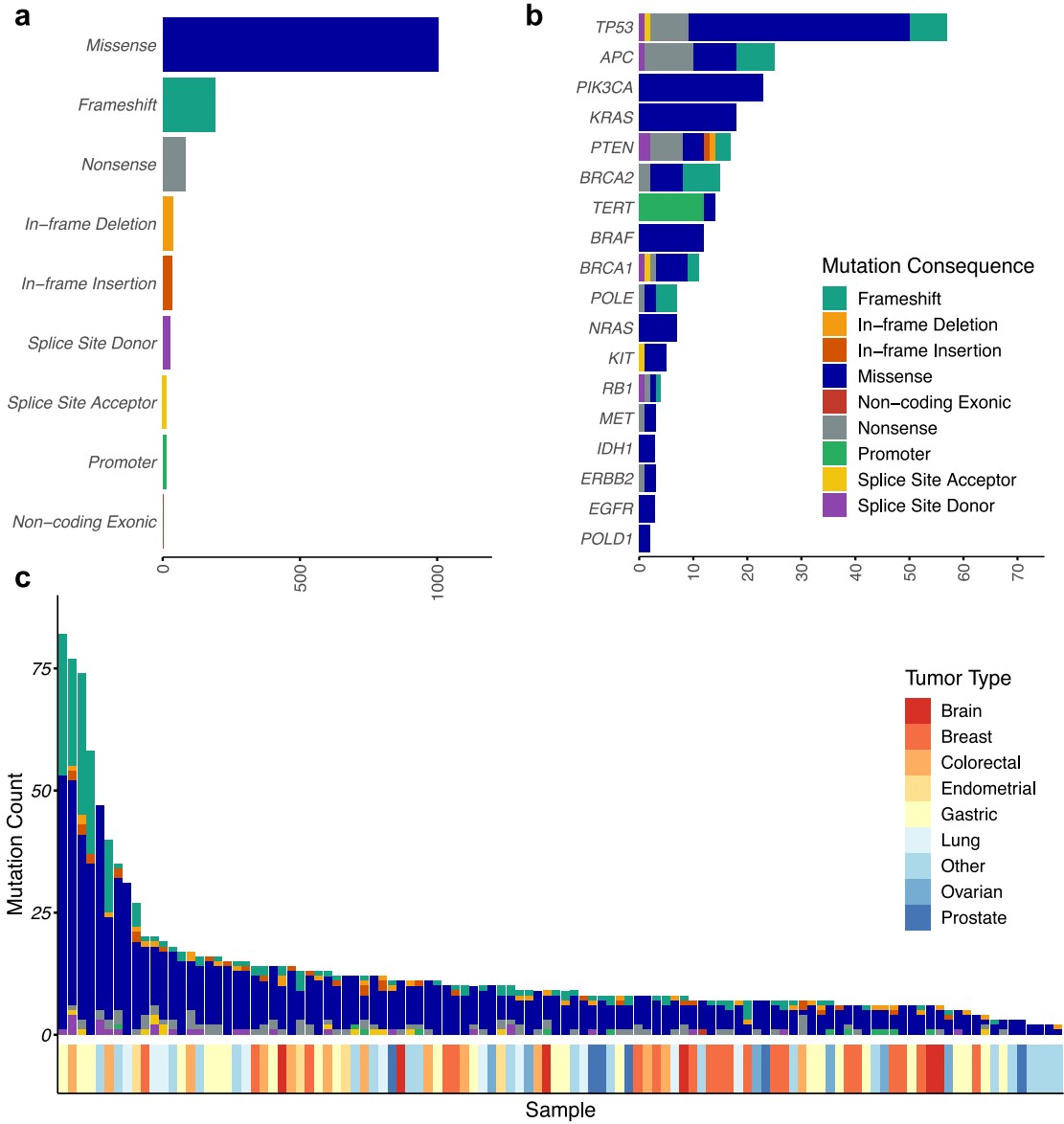

**Fig. 2 Genomic landscape of sequence variants identified by elio tissue complete.** Distribution of variant consequences identified among genes included in elio tissue complete across 112 FFPE tumor specimens analyzed (**a**), with specific alterations in select driver genes highlighted in (**b**). **c** The landscape of mutations identified, by consequence, per sample demonstrated a wide dynamic range in the number and type of variants identified across the cohort, with the tumor types indicated below each case. Two cases, in which no alterations were identified, are not displayed. Source data are provided as a Source Data file.

Data 3). We first analyzed performance across the 36 identified variants that were considered to have strong clinical significance (Supplementary Data 4). Among these samples, we also identified 35 of the 36 single nucleotide variants (SNVs) detected in the alternate assay, resulting in a positive percent agreement (PPA) of 97.2%. The variant that was not reported, *KRAS* K117N, was confirmed as present by ddPCR at 1.09% MAF, which was below the reporting threshold for this mutation in our variant calling model. Two SNVs of strong clinical significance were identified using elio tissue complete in each of the cohorts but not by the MSK-IMPACT or FoundationOne assays, resulting in a negative percent agreement of 99.93% and 99.95% to MSK-IMPACT and FoundationOne, respectively, and 99.94% overall. Both discrepant variants were truncating splice site or nonsense alterations detected in *BRCA1*, where there are known differences in the limit of detection and in germline variant reporting between these assays (Supplementary Data 5 and 6).

High agreement between tests was also observed for SNVs or short sequence insertions or deletions (1–40 bp) at somatic hotspots with potential clinical significance, as well as for non-hotspot variants in the genes throughout the panel. Of the hotspot SNVs and indels included in this set, 116/119 alterations were identified by elio tissue complete resulting in an overall hotspot PPA of 97.5%. Seven hotspot variants reported only in elio tissue complete contributed to an observed NPA >99.9%. When all alterations were included, we observed a PPA of 85% (727 observed alterations of 855 detected by orthogonal assays) and an NPA of >99% (Supplementary Data 5.4 and 5.5). Of the 128 discrepant single base mutations identified by the FoundationOne or MSK-IMPACT assays, 106 were detected but not reported by elio tissue complete, due to either low variant quality as scored by PGDx Cerebro, classification as germline, annotated as alternate gene transcripts, or proximity to low-complexity regions (Supplementary Data 6).

We also measured the accuracy of elio tissue complete against two PCR-based COBAS assays and one ddPCR-based assay for specific SNVs and indels in 58 FFPE tumors (Supplementary Data 5). In 34 samples tested for *BRAF* alterations, 12 were observed to have *BRAF* V600 variants while 22 had wild-type sequences, resulting in a 100% PPA between elio tissue complete and the orthogonal assay. In 56 FFPE tumor samples tested for changes in the *EGFR* gene, 26 had *EGFR* alterations (10 confirmed sequence alterations and 15 confirmed EGFR exon 19 deletions) while 30 were wildtype, resulting in a 100% PPA between the assays. Overall, the comparison resulted in a 100% NPA between elio tissue complete and at least one of the orthogonal PCR-based assays.

To establish the limit of detection (LoD) of elio tissue complete, three cell-line blends with clinically significant variants in *EGFR*, *KRAS*, *NRAS*, *BRCA1*, and *BRCA2* were mixed with wild-type DNA to generate five dilution levels. For each dilution level, 10 replicates were analyzed with elio tissue complete providing a total of 150 observations. The LoDs were determined as the average MAF at the lowest dilution level with 100% detection across the ten replicates. These analyses demonstrated the ability of elio tissue complete to detect variants of interest at MAF levels of 2–4% (Supplementary Data 7). To validate these observed LoDs, we analyzed six FFPE clinical cases, each having one variant of strong clinical significance. Each case was diluted with noncancerous, FFPE-derived DNA so the MAF of the variant would be near the expected limit of detection established in the previous study, and 20 replicates of each case were analyzed through elio tissue complete. Through these subsequent studies, we confirmed that detection of these variants could be achieved in all 20 replicates (sensitivity of 100%) at MAFs of 3–6% (Supplementary Data 7).

Finally, to ensure the specificity of elio tissue complete, we analyzed 34 postmortem noncancerous FFPE samples for variants of strong or potential clinical significance. No variants of strong clinical significance, including in *EGFR*, *BRAF*, *BRCA1*, *BRCA2*, *KRAS*, and *NRAS* were identified across the set of samples analyzed. In addition, no hotspot indels of potential clinical significance were identified, although a *DNMT3A* R882H variant in one individual was identified at 9% in both replicates of one sample. This alteration is identical to hotspot changes observed in *DNMT3A* in individuals with clonal hematopoiesis of indeterminate potential, suggesting that this change may be a result of aberrant white blood cell proliferation in this individual[25]. Overall, these analyses resulted in 100% specificity for variants of strong clinical significance and >98% specificity for variants with potential clinical significance (Supplementary Data 8).

**Training of tumor mutation burden prediction algorithm in non-small cell lung cancer**. Using the 1.3-Mb panel designed from the previous TCGA analyses (Fig. 3a and Supplementary Fig. 1), we developed and trained a computational algorithm to accurately estimate TMB using the mutations identified by WES as the gold standard. We analyzed 95 NSCLC FFPE samples with tumor cellularity ≥20% as well as patient-matched normal blood samples. In addition, 11 lung cancer cell lines with patient-matched normal cell lines derived from peripheral blood B lymphoblasts were included in the training cohort and serve as potential reference standards. In total, 3006 Gb of sequence data were obtained from WES of the NSCLC FFPE tumor and patient-matched normal samples, corresponding to an average of 178-fold and 95-fold distinct coverage for each tumor and normal sample evaluated, respectively. In addition, 327 Gb of sequence data were generated from the lung cancer cell-line samples, corresponding to an average of 168-fold distinct coverage and 94-

fold distinct coverage for each tumor and normal sample analyzed, respectively (Supplementary Data 9). Using the tumor and matched-normal WES data, we identified somatic nonsynonymous single nucleotide variants, splice-site alterations, insertions, and deletions and determined exome-wide TMB for these samples.

The same 95 NSCLC samples and 11 lung cancer cell lines without the matching normal samples were also evaluated with elio tissue complete, generating 1116 Gb of sequence data, corresponding to an average 1222-fold distinct coverage and 1798-fold distinct coverage, respectively, for each tumor or cancer cell-line sample evaluated (Supplementary Data 3). Candidate somatic mutations were identified from the 1.3 Mb coding regions of the targeted panel using the PGDx Cerebro machine-learning approach[23], and variant filtering criteria and regression models were evaluated to optimize TMB estimation. Variant characteristics that were considered included the MAF of the sequence alteration, the type of alteration (synonymous, nonsynonymous, splice site, insertion, and deletion changes), and the prevalence of these in various databases, including dbSNP, COSMIC, ExAC, and gnomAD. The number of variants with different combinations of these characteristics were fit to a log-transformed linear regression model to optimally predict the TMB observed from WES analyses and the mutation burden was reported as the elio-predicted exome TMB (eTMB) in mutations per megabase sequenced (mutations/Mb, exome equivalent) (see "Methods"). The filtering and regression algorithm with the highest correlation resulted in an eTMB that accurately estimated the WES reference TMB using a cross-validated approach (Pearson $r = 0.909$, $P < 0.0001$, Fig. 3b). Overall, the median difference between eTMB and the reference exome TMB was within 1.5 mutations/Mb (Supplementary Fig. 3a).

**Validation of the analytical accuracy of eTMB in a pan-solid tumor setting**. To validate the performance of the TMB prediction algorithm, an independent cohort enriched for tumor types for which immunotherapy is under evaluation or has demonstrated clinical efficacy was evaluated using elio tissue complete and WES. A total of 307 FFPE-derived tumor samples, including bladder ($n = 23$), breast ($n = 22$), colorectal ($n = 30$), endometrial ($n = 22$), head and neck ($n = 4$), kidney ($n = 16$), lung (NSCLC, $n = 110$, Unspecified Type, $n = 24$), stomach ($n = 13$) cancer, as well as melanoma ($n = 43$) were evaluated through WES along with patient-matched normal samples. WES analyses led to a 137-fold distinct coverage on average for each tumor exome evaluated (Supplementary Data 9), while targeted analyses of the FFPE tumor samples corresponded to a 930-fold distinct coverage on average for each tumor evaluated (Supplementary Data 3). Across the tumors analyzed, the average number of mutations was approximately 9.3 mutations/Mb and ranged from <0.1 mutations/Mb to 284.4 mutations/Mb as determined through WES analyses (Fig. 3c). We observed high concordance between the predicted TMB and the WES TMB (Pearson $r = 0.949$, $P < 0.0001$, Fig. 3c, d). Candidate variants were identified in 495 genes in the targeted panel and ranged from 5 to 100% MAF (Supplementary Data 10). Overall, the eTMB score was within a median of 1.3 mutations/Mb, exome equivalent of the observed TMB score obtained from WES data in this cohort (Supplementary Fig. 3b).

**Validation of the tumor mutation burden algorithm analytical performance**. In order to evaluate the performance and robustness of elio tissue complete, we determined the minimum DNA amount and tumor purity that could be analyzed, the limit of blank, as well as the precision and repeatability of the assay across

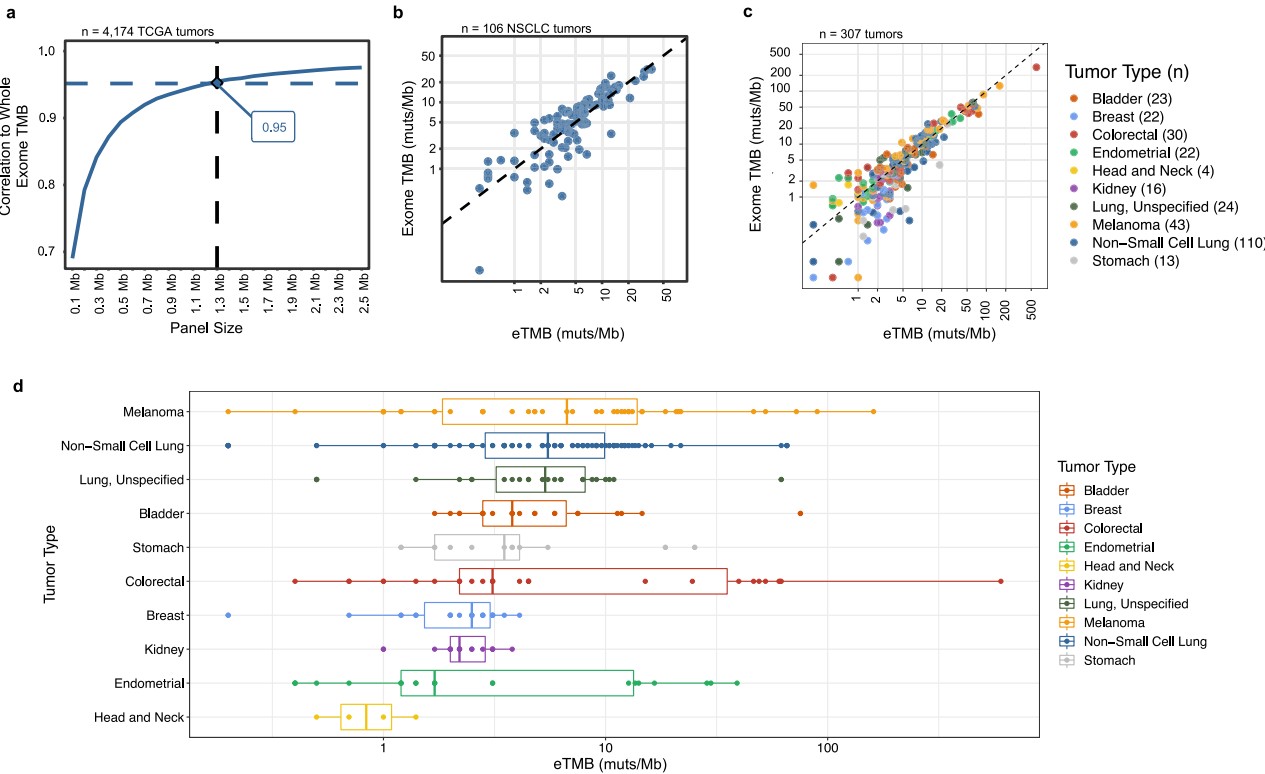

**Fig. 3 In silico and experimental comparison of the targeted panel and whole-exome TMB performance. a** The correlation of in silico predicted tumor mutation burden (TMB) for panels of different size (100 kilobases to 2.5 megabases, Mb) to observed whole-exome sequencing (WES) TMB for samples in The Cancer Genome Atlas (TCGA) suggested that panels of >1 Mb provide accurate TMB measurements. TMB analyses were performed in a metastatic population weighted according to the relative frequency of late-stage cases per year. **b** Comparison of the elio-predicted exome TMB (eTMB) using elio tissue complete and WES of tumor and matched-normal samples in a cohort of 106 non-small cell lung cancer (NSCLC) FFPE and cell-line samples resulted in high concordance in a cross-validated analysis (Pearson correlation = 0.909, $P < 0.0001$ and Spearman rho = 0.855, $P < 0.0001$). **c** Evaluation of eTMB in an independent cohort of 307 FFPE-derived pan tumor samples demonstrated high correlation to WES (Pearson correlation = 0.949, $P < 0.0001$ and Spearman rho = 0.870, $P < 0.0001$). **d** Distribution of eTMB scores in the independent cohort of 307 FFPE-derived tumors by tumor type, with the number of each tumor type captured in (**c**). The boxes represent the 25th and 75th percentile (interquartile range, IQR), while the median is reflected by the middle line of the box. The whiskers represent 1.5*IQR, with outliers plotted as points not connected to the whiskers. The Pearson and Spearman correlation coefficients were calculated using a two-sided test, and no adjustments were made for multiple comparisons. Source data are provided as a Source Data file. Muts/Mb mutations per megabase.

sites, instruments, operators, and days. To evaluate the limit of blank, or the apparent TMB in samples that would not be expected to have any somatic mutation burden, we analyzed 36 FFPE-derived samples obtained from postmortem noncancerous tissue. A detailed pathologic review confirmed the absence of tumor cells in these noncancerous samples. Twenty-two of these samples were processed in duplicate, resulting in 58 samples that were analyzed with a 653-fold distinct coverage on average (Supplementary Data 3). We observed an average of 0.9 mutations/Mb, exome equivalent in these normal tissue samples (Supplementary Fig. 4a). The limit of blank, established by calculating the 95th percentile of the eTMB across the 58 cases, was determined to be 1.9 mutations/Mb, exome equivalent (Supplementary Fig. 4a). The low-level rate of background sequence alterations may be a result of rare germline variants as well as artifacts due to DNA damage from formalin fixation, PCR or sequencing errors, or sequence misalignment. An additional set of 63 FFPE-derived samples obtained from postmortem non-cancerous tissue confirmed this LOB, as 95.2% (60/63) samples had an eTMB of ≤1.9. When the eTMB pan-solid tumor validation samples were reanalyzed to exclude samples with scores below the limit of blank, we continued to observe high concordance between eTMB and WES TMB (Pearson $r = 0.950$ $P < 0.0001$).

To confirm the minimum DNA input required to accurately estimate TMB, two lung cancer cell lines (NCI-H2087 and NCI-H2122) were diluted to 20% tumor purity using their matched-normal reference DNA and analyzed using elio tissue complete in triplicate with 50 ng, 75 ng, 100 ng, and 200 ng of total DNA. Using the mean eTMB score obtained from the 100 ng DNA input as a reference, we determined the percent error in eTMB for each replicate (Supplementary Fig. 4b). NCI-H2087 had a mean eTMB score of 11.5 mutations/Mb, exome equivalent with a median and maximum absolute percent deviation of the TMB score of 2.6% and 6.1% across all replicates. Similarly, NCI-H2122 had a mean eTMB score of 4.5 mutations/Mb, exome equivalent and no deviation in eTMB score across replicates. These data suggest that eTMB analyses are reliable even using limited DNA amounts from small tumor specimens.

To quantify the effect of tumor purity on the eTMB score, four tumor-derived cell lines (two NSCLC and two breast cancers) and 10 FFPE-derived tumor samples (seven NSCLC, two endometrial, one colorectal cancer) were obtained with a range of TMB scores (1.0–39.1 mutations/Mb, exome equivalent) as determined by elio tissue complete. The tumor purity of the FFPE-derived tumor samples ranged from 50 to 90%, as determined by pathologic review. For each sample evaluated, the undiluted sample and at least five serial dilutions were prepared using the

matched-normal-derived DNA as the diluent (Supplementary Data 3). To determine the minimum tumor purity that would result in accurate estimation of TMB, we calculated the tumor purity at which the mean eTMB at a dilution level deviated from the eTMB measured in the undiluted sample by >30% (see "Methods"). Of the 14 samples tested, nine (64%) deviated from the reference eTMB score by <30% when the diluted tumor purities were 20% or above, and all samples deviated by <30% when the diluted tumor purities were 35% or above (Supplementary Fig. 5).

We sought to understand if the five samples with >30% deviation in eTMB at tumor purities between 20% and 35% deviated at higher tumor purity due to a higher number of subclonal sequence alterations. We calculated the fraction of reads harboring a variant, corrected for tumor purity, as an estimate of tumor clonality by dividing the median sequence mutation MAF for the variants identified in the tumor by the pathologic tumor purity. While this measure of tumor clonality may have been influenced by imprecise estimates of pathological purity and only considers a subset of tumor variants, we observed a negative correlation between the tumor purity at which the eTMB score deviated by >30% from the undiluted score and the derived clonality score (Pearson correlation = −0.840, $P = 0.0002$) (Supplementary Fig. 6). These data suggest that higher sequence mutation clonality, and thus lower tumor heterogeneity, is associated with more accurate TMB scores at lower tumor purity.

To assess the precision and intra-run repeatability of the TMB score obtained from elio tissue complete across a variety of conditions, six contrived samples containing mixtures of cell lines, one unaltered cell line, and 14 FFPE cancer specimens were analyzed. Two replicates of each sample were each run at three independent sites with two operators per site, and with each operator preparing the two replicates on three different days and at least one instrument for a total of at least 33 replicates per sample and a total observation count of 829. High concordance between the reported results across replicates was observed as demonstrated by an overall coefficient of variation (CV) of <8% for all samples with an eTMB above the limit of blank (1.9 mutations/Mb, exome equivalent). Across all specimens, the highest CV observed among the samples analyzed within the same runs was 5.6%, 1.8% between days, 2.1% between sites, and 1.8% between operators (Fig. 4). In addition, the coverage achieved across replicates was highly reproducible, with overall CV ranging from 8.5 to 16.4% for total coverage and 12.6 to 26.3% for distinct coverage. These data confirm the precision and repeatability of elio tissue complete under normal operating conditions of a clinical laboratory, a critical requirement for a decentralized system to quantify TMB from FFPE tumor specimens.

To further characterize the performance of the eTMB algorithm, we compared 31 clinical FFPE specimens in an independent head-to-head comparison with the ThermoFisher Oncomine Tumor Mutation Load assay. The cohort was chosen to cover a range of TMB values as observed in tumor and matched-normal WES data analyzed with the Strelka2 variant caller (range 0.06–28.51 mutations/Mb). When compared to the TMB measured through WES of the same samples, the eTMB algorithm outperformed the ThermoFisher assay (Pearson $r = 0.926$ and 0.748 for elio tissue complete and ThermoFisher Oncomine Tumor Mutation Load, respectively, Supplementary Fig. 7). The elio tissue complete test displayed a wider dynamic range that more accurately reflected the WES TMB (range 0.4–30.4 mutations/Mb, exome equivalent) while the Thermo-Fisher assay had a more limited TMB dynamic range (range 6.7–68.1 mutations/Mb) that had a lower correlation to WES TMB.

**Development and training of the MSI detection algorithm**. To develop an NGS approach for the detection of MSI in FFPE tumor samples, we identified all mono-, di-, and trinucleotide repeat tracts across the elio tissue complete targeted region of interest. From the panel, candidate tracts were evaluated for their ability to distinguish microsatellite instability-high (MSI-H) from microsatellite stable (MSS) status in a total of 36 MSI-H and 96 MSS clinical samples analyzed with elio tissue complete that had MSI status confirmed through orthogonal methods, as well as 755 additional FFPE tumor specimens. We applied our previously described MSI peak finding analysis algorithm, modified for FFPE tumor tissue analyses, for 68 repeat tracts[26]. Individual tracts were classified as unstable if the allele length was ≥2 bp shorter than the reference length and samples were considered MSI-H if more than 10 of the 68 (16%) analyzed tracts were identified as unstable. This repeat tract-based approach resulted in a 100% positive and negative percent agreement with the independently obtained MSI status of the 132 cases in our training set.

As tract-based approaches may have limitations in identifying MSI tumors due to variability in the effect of mismatch repair (MMR) deficiency on various tracts as well as the presence of germline changes in repeat sequences in some individuals, we developed a mutation-based method that could independently determine MSI status in such cases. We developed a position weight matrix (PWM) model to represent the contextual mutation signatures associated with MMR deficiency[27]. We trained the PWM model through analysis of exome sequences from >2500 tumor samples that had been independently evaluated for MSI[28]. The model reports the log-likelihood ratio of a substitution coming from an MMR-deficient tumor vs. an MMR-proficient tumor. Because MSI is caused by MMR deficiency, mutations in an MSI tumor will tend to have positive numbers reported by the model, and mutations from an MSS tumor will tend to have negative numbers reported by the model. For each tumor, we calculated an MSI signature score by evaluating each substitution with our model and determining the sum of all substitution scores. We evaluated the performance of the approach using exome sequences from 78 MSI-H and 186 MSS tumor samples and observed a 95% PPA and 100% NPA with the MSI status of these samples as determined by PCR analyses (Supplementary Fig. 8). Importantly, this method does not appear to be affected by other DNA repair deficiencies or mutagenic processes, including *POLE* and *POLD1* hypermutators, and alterations resulting from UV radiation (Supplementary Fig. 8).

We combined the signature score with our tract-based score to form an ensemble MSI classifier. To train this classifier, we used 725 cancer samples, representing a combination of unique cases and technical replicates, with 73 (of the previous 132 cases) containing confirmed MSI status. Using these scores plotted on a two-dimensional plane (Fig. 5a), we established a linear decision boundary line that separated the MSI-H and MSS cases, while maintaining a roughly equal distance between the boundary and nearest MSI-H and MSS points, to develop a combined score.

**Validation of the MSI detection algorithm analytical performance.** To assess the accuracy of the combined MSI algorithm, 223 clinical FFPE specimens from a pan-solid tumor cohort including colorectal ($n = 66$), stomach ($n = 27$), lung ($n = 19$), and endometrial ($n = 18$) cancers were evaluated using elio tissue complete. In total, 2232 Gb of sequencing data was generated for this study and 972-fold distinct coverage was obtained, on average, for each tumor (Supplementary Data 3). Of the 80 MSI-H samples identified by multiplex PCR, 79/80 samples (98.8%) were identified as MSI-H by the combined tract-level and signature

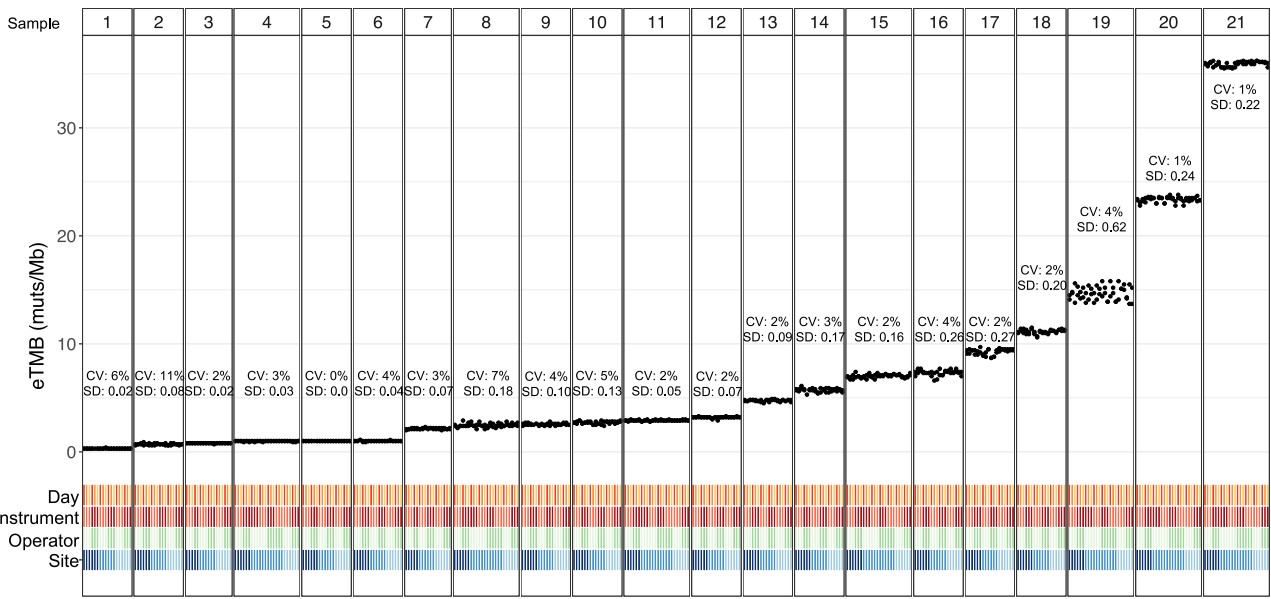

**Fig. 4 Analytical precision and reproducibility of elio tissue complete eTMB.** Precision and reproducibility of elio-predicted exome tumor mutation burden (eTMB) measured in mutations per megabase (muts/Mb) across three independent clinical laboratories using 21 FFPE tumor and cell-line samples. At each site, each sample was analyzed by two operators on two sequencing instruments across three non-consecutive days as indicated. The coefficient of variation (CV) and standard deviation (SD) for each sample are indicated, demonstrating high performance of eTMB across standard clinical laboratory variables. Source data are provided as a Source Data file.

MSI prediction method (Fig. 5b), including in colorectal ($n = 35$), endometrial ($n = 16$), gastric ($n = 8$), stomach ($n = 8$), prostate ($n = 3$), lung ($n = 1$), ovarian ($n = 1$), thyroid ($n = 1$), and other ($n = 6$) cancers. The one case that was not detected as MSI-H by elio tissue complete had 3/5 tracts positive by multiplex PCR and each altered tract was noted to have only a subtle size shift. In addition, of the 143 MSS samples identified by multiplex PCR, 99.3% (142/143) of samples were scored as MSS by our combined method. The additional case we identified as MSI-H had a one base pair (bp) insertion in *MSH6* that has been reported as pathogenic (NCBI ClinVar Accession VCV000141667.3). Importantly, four of the analyzed samples would have been incorrectly categorized by the tract-based approach alone but were correctly classified using the combined analysis (Fig. 5b). Similar to the training cohort, the combined method was not influenced by hypermutators driven by other DNA repair deficiencies or mutagenic processes (Supplementary Fig. 9). Interestingly, only a subset of the repeat tracts (29 of 68) had perfect specificity in MSS samples, and none had perfect sensitivity in the MSI-H samples (Fig. 5c). These data suggest that our combination approach using repeat tracts and mutation signatures in elio tissue complete can accurately classify MSI status across a broad range of mismatch repair-deficient tumors.

To determine whether MSI status may be affected by different levels of tumor purity, an FFPE specimen with confirmed MSI status was diluted with its matched-normal DNA to simulate five tumor purities ranging from 37 to 18%. Ten replicates of each tumor purity level were prepared and analyzed using elio tissue complete. MSI-H was detected in all 50 samples (100% sensitivity), indicating that MSI status was not affected above 20% tumor purity. To confirm this claim, three additional MSI-H FFPE specimens of varying tumor purities were diluted with their matched-normal DNA to tumor purities from 20 to 15%. All 40 replicates for each sample above 16% tumor purity were correctly identified as MSI-H, while among the replicates at 15% tumor purity, 19 of 20 were identified as MSI-H. These data confirm the

sensitivity of the elio tissue complete MSI algorithm for specimens with >15% tumor purity.

Finally, to evaluate the repeatability and precision of the MSI algorithm in a decentralized laboratory environment, six contrived samples containing mixtures of cell lines (three MSI-H and three MSS), one unaltered cell line (MSS), seven MSI-H FFPE cancer specimens, and seven MSS FFPE cancer specimens were analyzed. Two replicates of each sample were each run at three independent sites with two operators per site, and with each operator preparing the two replicates on three different days and at least one instrument for a total of at least 33 passing replicates per sample and a total observation count of 829. Average positive and average negative agreements were calculated to be 99.1% and 99.3%, respectively, across all comparisons between sites, operators, and days (Fig. 6), highlighting the performance of the elio tissue complete test across independent laboratories with trained operators to precisely and repeatedly produce accurate MSI results.

**Validation of structural variant detection analytical accuracy.** To develop an approach for the detection of structural variants, we included select intronic regions for detection of translocations in 12 genes and additional heterozygous SNPs for detection of gene amplifications in 16 genes. We adapted the previously described Digital Karyotyping (DK)[29] and Personalized Analysis of Rearranged Ends (PARE)[29] algorithms to the elio tissue complete panel using FFPE-derived tumor specimens and cell lines with known positive variants as well as FFPE-derived non-cancerous specimens for training. After completing algorithm optimization, we analyzed >340 FFPE-derived tumor specimens with elio tissue complete and compared the results against DNA- and RNA-based orthogonal assays. Overall, we achieved 86.4% (121/140) PPA and 98.8% (2106/2132) NPA for gene amplification compared to orthogonal NGS and fluorescence in situ hybridization (FISH)-based assays as well as 82.4% (42/51) PPA

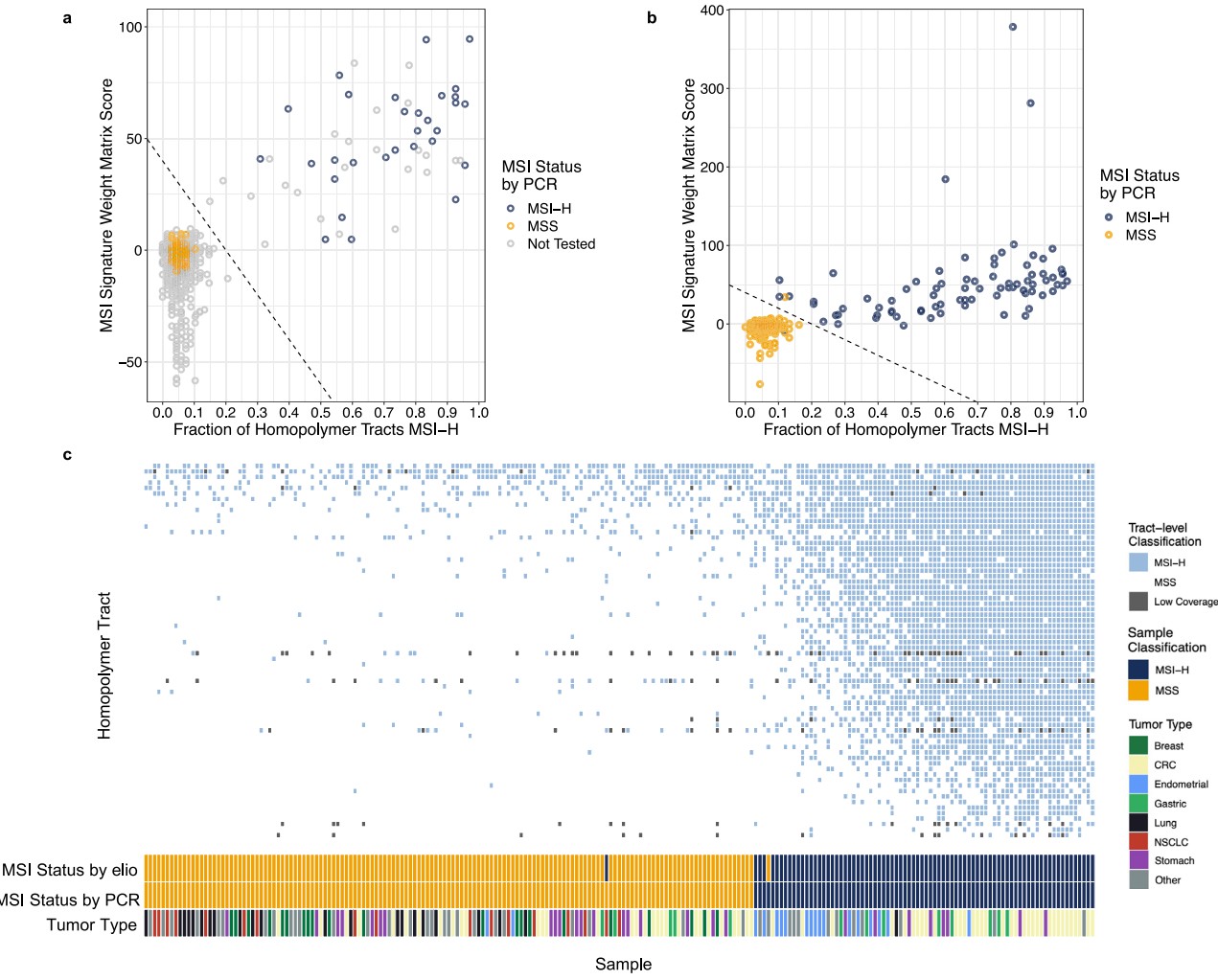

**Fig. 5 Training and analytical validation of the elio tissue complete MSI detection algorithm. a** 725 FFPE cancer samples were evaluated using the combination tract-based and substitution score microsatellite instability (MSI) algorithm. A subset of these samples ($n = 73$) had confirmed MSI status through multiplex PCR analyses. After plotting both the fraction of positive homopolymer tracts and the signature weight matrix score, a decision boundary was determined to separate the cluster of known microsatellite stable (MSS) cases from known microsatellite instability-high (MSI-H) cases. Analysis for the presence or absence of deleterious mutations in genes involved in mismatch repair was used to segregate the population of samples close to the decision boundary line. **b** An independent cohort of 223 FFPE cancer samples with confirmed MSI status was analyzed with the elio tissue complete MSI algorithm. **c** Detailed analyses of each of the 68 mononucleotide tracts employed in the tract-based peak finding algorithm demonstrated >40% of tracts have perfect specificity in MSS cases and that a combination of high-sensitivity and high-specificity tracts was employed in the peak finding algorithm. Source data are provided as a Source Data file. CRC colorectal cancer, NSCLC non-small cell lung cancer.

and 99.9% (1220/1221) NPA for translocations compared to independent DNA- and RNA-based approaches (Table 1). Notably, elio tissue complete was able to detect 92.9% (13/14) of *ALK* translocations and 87.0% (40/46) of *ERBB2* amplifications (Supplementary Fig. 10) when compared to FISH. Taken together, these data demonstrated the high accuracy of elio tissue complete compared to standard reference assays for the detection of structural variants.

## Discussion

The connection between mutations in the genome and neoplastic transformation has been well established, and therapies targeting tumor-specific genetic abnormalities, both large structural changes and small activating-sequence mutations, have proven effective across a range of cancer types. Identification of patients with these targeted genetic markers is imperative for effective treatment, and, historically, single-analyte diagnostic tests have been instrumental in directing patient care with these targeted

therapies. While NGS targeted panels may not always be necessary to assess a small number of actionable or prognostic targets used for therapy selection in certain clinical settings, they do allow for more comprehensive testing of all current and potential future biomarkers from a single sample preparation. Based on comparisons to orthogonal data from two FDA-cleared NGS-based tumor profiling assays, elio tissue complete has demonstrated high analytical performance for both variants associated with therapies or clinical decisions with a PPA of 97.2% and NPA of >99% and variants with potential biomarker significance with a PPA of 85%. These variants can be detected to biologically relevant levels of 3–6%, below the expected allele fraction for a clonal variant in most tumors.

In addition to targeted therapies, recent developments in immunotherapy treatment, such as the use of checkpoint inhibitors for the management of multiple types of cancers, have demonstrated durable clinical responses. Pembrolizumab, a PD-1 inhibitor, was approved for MSI-high patients, becoming the first biomarker to receive drug approval regardless of cancer type[8].

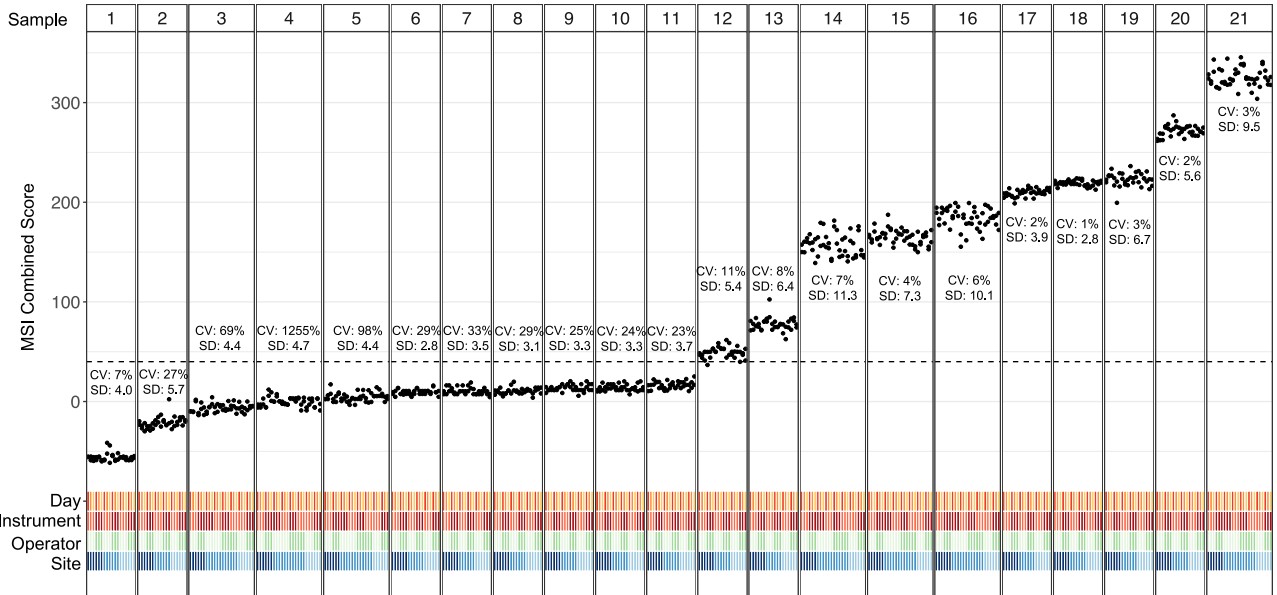

**Fig. 6 Analytical precision and reproducibility of the elio tissue complete MSI detection algorithm.** Precision and reproducibility of the microsatellite instability (MSI) status from elio tissue complete were evaluated across three independent clinical laboratories using 21 FFPE and cell-line samples. At each site, each sample was analyzed by two operators on two sequencing instruments across three non-consecutive days as indicated. The absolute value of the coefficient of variation (CV) and standard deviation (SD) for each sample are indicated. Samples with a combined score above the dotted line are considered microsatellite instability-high, demonstrating high performance of MSI across standard clinical laboratory variables. Source data are provided as a Source Data file.

**Table 1 Summary of amplification and translocation detection analytical performance.**

| Gene | Variant type | PPA (%) (n/N) | NPA (%) (n/N) | Orthogonal assay | Reference | Analysis mode |
|---|---|---|---|---|---|---|
| Aggregate (n = 12 genes) | Translocation | 82.4% (42/51) | 99.9% (1220/1221) | N/A | N/A | N/A |
| Aggregate (n = 16 genes) | Amplification | 86.4% (121/140) | 98.8% (2106/2132) | N/A | N/A | N/A |
| ALK | Translocation | 92.9% (13/14) | 98.2% (56/57) | Vysis ALK Break-Apart FISH Probe | This study | IVD and RUO |
| BRAF | Translocation | 100% (2/2) | 100% (145/145) | FoundationOne | Deak et al.[24] | RUO |
| CCND1 | Amplification | 72.7% (8/11) | 99.3% (146/147) | FoundationOne | Deak et al.[24] | RUO |
| CCND2 | Amplification | 75.0% (3/4) | 99.3% (146/147) | FoundationOne | Deak et al.[24] | RUO |
| CCND3 | Amplification | 50.0% (1/2) | 100% (147/147) | FoundationOne | Deak et al.[24] | RUO |
| CCNE1 | Amplification | 92.9% (13/14) | 98.6% (145/147) | FoundationOne | Deak et al.[24] | RUO |
| CD274 | Amplification | 100% (2/2) | 98.6% (145/147) | FoundationOne | Deak et al.[24] | RUO |
| CDK4 | Amplification | 66.7% (2/3) | 100% (147/147) | FoundationOne | Deak et al.[24] | RUO |
| EGFR | Amplification | 100% (7/7) | 99.3% (146/147) | FoundationOne | Deak et al.[24] | RUO |
| EGFR | Translocation | 50% (1/2) | 100% (147/147) | FoundationOne | Deak et al.[24] | RUO |
| ERBB2 | Amplification | 87.0% (40/46) | 95.9% (71/74) | LSI HER2/neu FISH Probe | This study | IVD and RUO |
| EWSR1 | Translocation | 100% (2/2) | 100% (147/147) | FoundationOne | Deak et al.[24] | RUO |
| FGFR1 | Amplification | 100% (11/11) | 100% (147/147) | FoundationOne | Deak et al.[24] | RUO |
| FGFR1 | Translocation | 100% (1/1) | 100% (147/147) | FoundationOne | Deak et al.[24] | RUO |
| FGFR2 | Amplification | 100% (1/1) | 99.3% (146/147) | FoundationOne | Deak et al.[24] | RUO |
| FGFR3 | Translocation | 100% (1/1) | 100% (147/147) | FoundationOne | Deak et al.[24] | RUO |
| MDM2 | Amplification | 88.3% (5/6) | 98.0% (144/147) | FoundationOne | Deak et al.[24] | RUO |
| MET | Amplification | 100% (5/5) | 97.3% (143/147) | FoundationOne | Deak et al.[24] | RUO |
| MYC | Amplification | 76.2% (16/21) | 95.2% (140/147) | FoundationOne | Deak et al.[24] | RUO |
| MYCN | Amplification | 100% (4/4) | 99.3% (146/147) | FoundationOne | Deak et al.[24] | RUO |
| NTRK1 | Translocation | 75.0% (3/4) | 100% (147/147) | FoundationOne | Deak et al.[24] | RUO |
| NTRK2 | Translocation | 100% (1/1) | 100% (69/69) | FoundationOne | This study | IVD and RUO |
| NTRK3 | Translocation | 66.7% (2/3) | 100% (12/12) | Archer Solid Tumor FusionPlex | This study | IVD and RUO |
| PDGFRA | Amplification | 100% (1/1) | N/A | FoundationOne | This study | RUO |
| PIK3CA | Amplification | 100% (2/2) | 100% (147/147) | FoundationOne | Deak et al.[24] | RUO |
| RET | Translocation | 55.6% (5/9) | 100% (18/18) | Vysis 10q11 RET Break-Apart FISH | This study | IVD and RUO |
| ROS1 | Translocation | 100% (1/1) | 100% (36/36) | Archer Solid Tumor FusionPlex | This study | RUO |
| TMPRSS2 | Translocation | 90.9% (10/11) | 100% (147/147) | FoundationOne | Deak et al.[24] | RUO |

Initially, MSI detection involved measuring five distinct micro-satellite regions (Bethesda-NCI Panel)[30], but advances in NGS have made it feasible to assess more regions, potentially improving the sensitivity and specificity for measuring MSI accurately[31,32]. The elio tissue complete test, including 68 repeat tracts and a signature score assessment, demonstrated high analytical concordance with an orthogonal method across a variety of indications and tumor purities, as well as high reproducibility across sites, operators, days, and sequencing instruments (>99% average PPA and NPA).

Although MSI has been an effective biomarker in several indications, such as colorectal and gastric cancers, other indications exhibited notably low rates of MSI[33]. Recently, immune checkpoint blockade has been approved for the treatment of adult and pediatric patients with unresectable or metastatic TMB-high solid tumors that have progressed following prior treatment and who have no satisfactory alternative treatment options. Acting as a surrogate for neoantigen load, TMB has been shown to predict response to immune checkpoint blockade[33]. Several studies have explored the impact of NGS panel size in calculating TMB. Smaller panels have been shown to be inaccurate in the assessment of TMB, with accuracy and precision increasing with larger panels[34–37]. Recent data suggests that a strong correlation to WES TMB necessitates a panel of at least 1 Mb, but that increasing panel size greater than 1.5 Mb confers little added benefit[20,35]. Through validation of our targeted approach, we verified high analytical concordance between the predicted eTMB and the whole-exome TMB for 307 pan-solid tumor specimens. The high analytical performance of elio tissue complete across a broad range of tumor types suggests that this approach would be useful in evaluating the clinical utility of proposed TMB thresholds (e.g., 10 mutations/Mb[37–40]) in future clinical trials.

A significant barrier to clinical adoption of NGS analyses is the lack of accessibility to validated platforms in local laboratories. Most validated methods are offered as send-out services, significantly increasing the turn-around time for results and may not be an option for all clinicians, such as those based internationally. In addition, differences in sequencing instruments and standardization for targeted panels can give rise to variation in TMB determination, making objective assessment problematic[41]. We have demonstrated through these analytical studies that elio tissue complete is highly specific, accurate, and reproducible for the measurement of SNVs, indels, amplifications, translocations, TMB, and MSI. The combination of the high-performance bioinformatics analyses and a kitted approach to sample preparation with elio tissue complete allows for a standardized evaluation of biomarkers using a comprehensive genomic profiling test. This potential for highly accurate, standardized results in a decentralized testing environment can enable more cancer patients to have access to this broad-range testing for their tumor's specific genetic mutations, improving patient access to more effective precision oncology treatment strategies.

## Methods

**Study population**. All patients provided written informed consent and the studies were performed according to the Declaration of Helsinki. Formalin-fixed paraffin-embedded (FFPE) tumor specimens were obtained under Institutional Review Board approval from Duke University (Pro00091621) and the National Cancer Institute (National Institutes of Health Clinical Center), as well as through commercial sources, including BioIVT (Hicksville, NY, USA), Indivumed (Hamburg, Germany), iSpecimen (Lexington, MA, USA), Folio Biosciences (Powell, OH, USA), Cureline (Brisbane, CA, USA), ProteoGenex (Inglewood, CA, USA), and Pathgroup (Brentwood, TN, USA). FFPE samples from noncancerous tissue were procured through Cureline. Human tumor and normal cells from previously characterized cell lines were obtained from ATCC (Manassas, VA, USA) (NCI-H1770 [CRL-5893]/NCI-BL1770[CRL-5960], NCI-H1672[CRL-5886]/NCI-BL1672[CRL-5959], NCI-H1395[CRL-5868]/NCI-BL1395[CRL-5957], NCI-H1437[CRL-5872]/NCI-BL1437[CRL-5958], NCI-H2009[CRL-5911]/NCI-

BL2009[CRL-5961], NCI-H2087[CRL-5922]/NCI-BL2087[CRL-5965], NCI-H2122[CRL-5985]/NCI-BL2122[CRL-5967], NCI-H2126[CCL-256]/NCI-BL2126[CCL-256.1D], NCI-H1184[CRL-5858]/NCI-BL1184[CRL-5949], NCI-H2171[CRL-5929]/NCI-BL2171[CRL-5969], NCI-H128[HTB-120]/NCI-BL128[CRL-5947], HCC1008[CRL-2320], HCC1937[CRL-2336], NCI-H1975[CRL-5908], HCC1954[CRL-2338]/HCC1954BL[CRL-2339], DLD-1[CCL-221], CHP-212[CRL-2273], NCI-H1650[CRL-5883], BT-474[HTB-20], and HCC1143BL[CRL-2362]) and Horizon Discovery (Waterbeach, UK) (HD753 and HD768).

**PGDx elio tissue complete test intended use and run modes**. The elio tissue complete test has been FDA-cleared as a qualitative in vitro diagnostic device that uses targeted next-generation sequencing of DNA isolated from formalin-fixed, paraffin-embedded tumor tissue from patients with solid malignant neoplasms to detect tumor gene alterations in a broad multigene panel. PGDx elio tissue complete is intended to provide tumor mutation profiling information on somatic alterations (SNVs, small insertions and deletions, one amplification [*ERBB2*] and four translocations [*ALK*, *NTRK2*, *NTRK3*, and *RET*]), microsatellite instability (MSI) and tumor mutation burden (TMB) for use by qualified healthcare professionals in accordance with professional guidelines in oncology for previously diagnosed cancer patients, and is not conclusive or prescriptive for labeled use of any specific therapeutic product. The results presented in this study represent the results obtained when analyzing these datasets in IVD-mode, and have been additionally reanalyzed in RUO-mode to provide data for specific amplifications and translocations that are not within the previously described intended use (Table 1).

**In silico TMB panel evaluation**. To evaluate the expected performance for TMB panels of variable size, somatic mutation data from the TCGA MC3 project (v0.2.8) were obtained from the Synapse repository (syn7214402 [https://www.synapse.org/#!Synapse:syn7214402/wiki/405297])[42]. Nonsynonymous mutations with mutant allele frequencies ≥10% and classified PASS were analyzed from the following cancer types: lung, colorectal, melanoma, bladder, uterine/endometrial, kidney, head and neck, liver, and gastric ($n = 4174$ unique samples). Panels of sizes 100 kb to 2.5 Mb were generated through random selection and assembly of exon ROIs. Mutations falling within each simulated panel ROI coordinates were counted as observed.

Calculation of performance for the general metastatic cancer population was performed by reweighting cancer-specific results based on relative new cases of "distant" cancer cases per year estimated from recent epidemiological studies and the SEER database (https://seer.cancer.gov)[22]. This reweighting of performance was as follows: lung (64.3%), colorectal (15.0%), melanoma (1.8%), bladder (1.6%), head and neck (5.9%), liver (3.7%), gastric (2.6%), and uterine/endometrial (5.1%).

**FFPE tumor and normal exome analyses**. Sample processing from tissue or buffy coat, library preparation, hybrid capture, and sequencing were performed at Personal Genome Diagnostics Inc. (Baltimore, MD)[15]. Briefly, DNA was extracted from FFPE tissue and matched-normal buffy coat cells using the Qiagen FFPE Tissue Kit and DNA Blood Mini Kit, respectively (Qiagen, Hilden, Germany; catalog numbers 56404 and 51104, respectively). Genomic DNA was sheared using a Covaris sonicator (Woburn, MA, USA) to a size range of 150–450 bp, and subsequently used to generate a genomic library using the New England Biolabs (Ipswich, MA, USA) end-repair, A-tailing, and adapter ligation modules (catalog numbers E6050, E6053, and E6056, respectively). Finally, genomic libraries were amplified and captured using the Agilent SureSelect XT in-solution hybrid capture system with a 120 bp RNA panel targeting the pre-defined regions of interest across full exonic regions. Captured libraries were sequenced on the Illumina HiSeq 2000 or 2500 (Illumina, San Diego, CA, USA) with 100-bp paired-end reads.

**FFPE tumor-targeted analyses**. Sample processing from tissue, library preparation, hybrid capture, and sequencing were performed at Personal Genome Diagnostics Inc. (Baltimore, MD)[15]. Briefly, DNA was extracted from FFPE tissue using the Qiagen FFPE Tissue Kit (Qiagen, Hilden, Germany; catalog number 56404). Genomic libraries were prepared, amplified, and captured using PGDx elio tissue complete sample preparation kits. Captured libraries were sequenced on the Illumina NextSeq 500 or 550 (Illumina, San Diego, CA, USA) with 150 bp paired-end reads.

**Measures of coverage for elio tissue complete**. Total coverage was calculated as the average total number of reads sequenced across the regions of interest in the elio tissue complete targeted panel. The distinct coverage was calculated as the average number of unique reads sequenced across the regions of interest in the elio tissue complete targeted panel. The variability observed in coverage across the cohort was a reflection of the pre-analytical quality of the samples analyzed in the study due to factors such as formalin fixation time/process, specimen age/quality, and/or biopsy type. The elio tissue complete test does not require a sample to pass pre-analytical quality requirements such as DNA integrity, but rather any sample that meets the minimum DNA input yield was processed and was evaluated against coverage quality metrics post-sequencing.

**Training of the PGDx Cerebro model for elio tissue complete**. The PGDx Cerebro machine-learning approach to variant identification uses an extremely randomized trees (or "Extra-Trees") classification model trained with sequence data sourced from noncancerous FFPE-derived DNA and synthetic somatic variants spiked-in to the aligned sequence files[23]. For the elio tissue complete training set, fourteen FFPE tumor-adjacent noncancerous samples from various ethnic origins were sequenced, and synthetic variants, of which 37,659 were SNVs, 43,056 were insertions, and 43,332 were deletions, were spiked into the aligned sequence files. The composition of the synthetic variants followed similar methodology as previously published[23] with adaptations to the spiked MAF range for a smaller targeted panel with higher sequencing depth. Synthetic variants were spiked-in at MAFs ranging from 0.46 to 99.97%, with >70% of variants spiked-in at clinically meaningful levels between 1 and 45% MAF. Indels of sizes ranging between 1 and 3 bp constituted roughly 66% of spiked indels, while challenging indels >15 bp in length and those occurring at polyN tracts were spiked-in at 20% and 14%, respectively. Unlike the exome model, an unmatched synthetic normal resulting from an in silico combination of an additional 16 FFPE tumor-adjacent non-cancerous samples was used as a normal sample for training. The Extra-Trees model was fit to this training data, resulting in a model optimized for elio tissue complete and tumor-only somatic variant identification.

**Candidate variant identification for whole-exome and targeted sequencing**. Next-generation sequencing whole-exome data were processed and variants were identified using the VariantDx custom variant calling software [v9][23]. Briefly, sequencing reads were aligned to the hg19 human reference genome using ELAND (v1.8.2; www.illumina.com) and Novoalign (v3.2.7; www.novocraft.com) aligners. After variants were identified by VariantDx, a set of optimized filters were applied to obtain a high-confidence set of variant calls. Variants with potential mismapping to the genome were removed, as well as variants with low evidence in the tumor (<10% mutant allele fraction). Variants with high frequency in the population as annotated by dbSNP (v138, https://www.ncbi.nlm.nih.gov/projects/SNP/snp_summary.cgi?view+summary=view+summary&build_id=138)[43] were removed as well as variants identified in the matched normal. A trained scoring algorithm further curated the list of candidate variants to return a high-confidence tumor mutation burden from the exome.

Next-generation sequencing targeted data were processed and high-quality variant calls were identified using the VariantDx custom variant calling software [v10.4] and PGDx Cerebro scoring algorithm[23]. Briefly, sequencing reads were aligned to the hg19 human reference genome using BWA-MEM [v0.7.15][44] and Bowtie2 [v2.3.1][45] aligners (Fig. 1). Variant calls were identified by VariantDx and assigned a confidence score by PGDx Cerebro [v20], taking into consideration the quality of sequencing reads and mapping. After removing variants with low confidence PGDx Cerebro scores, putative germline variants were removed if they had high frequency in the population as annotated by dbSNP (v138, https://www.ncbi.nlm.nih.gov/projects/SNP/snp_summary.cgi?view+summary=view+summary&build_id=138)[43], ExAC (v1, https://gnomad.broadinstitute.org/downloads#exac-variants)[46] and gnomAD (v2.0.2, https://gnomad.broadinstitute.org/downloads)[47]. Additional filters based on mapping quality and genomic complexity ensure variants were not called due to artefactual processes. Finally, reported variants must have yielded either 4 or 6 variant observations and a MAF above 0.4–5.0%, depending on the level of evidence for clinical actionability and cancer driver potential. All deleterious *BRCA1* and *BRCA2* variants were reported if the number of mutation observations and allele fraction passed thresholds, regardless of whether they were considered germline or somatic.

**Orthogonal testing for validation of sequence variant identification**. To evaluate the accuracy of the sequence variant identification algorithm, 112 tumor FFPE samples were tested by one of two orthogonal NGS assays. Seventy samples were tested by both elio tissue complete and the FoundationOne (Foundation Medicine, Inc.) assay using standard protocols at the time of processing[24]. Only data describing the affected gene and resulting amino acid change were available from the FoundationOne assay. Many variants with high population frequency reported in dbSNP[43], ExAC[46], or gnomAD[47] were reported by FoundationOne but not by elio tissue complete. Where elio tissue complete detected but did not report a variant due to the high population frequency, the variant was removed from analysis on the basis of reporting differences. The additional 42 samples were analyzed by the MSK-IMPACT assay (Memorial Sloan Kettering Cancer Center) using standard protocols. Genomic coordinates of the identified variants were available for this cohort, and analysis was performed based on these genomic alterations rather than gene and amino acid annotation. The MSK-IMPACT assay has also published blacklisted regions of interest, and variants identified in those regions were removed from analysis. Finally, all variant calls in genes not analyzed by both assay panels were removed from analysis.

In addition to the two NGS-based assays described above, three PCR-based assays were used to confirm the accuracy of selected *BRAF* and *EGFR* variants. Two tests, the COBAS 4800 BRAF V600 Mutation Test and COBAS EGFR Mutation Test v2 test (Roche Diagnostics), identified variants in *BRAF* V600, *EGFR* T790M, *EGFR* L858R, and deletions in *EGFR* exon 19. An additional ddPCR test validated

internally at Personal Genome Diagnostics, Inc. was used for discrepancy analysis between elio tissue complete and the COBAS tests.

Performance was reported in categories of strong clinical significance and potential clinical significance based on FDA guidance[48]. Hotspot variants were determined based on common cancer driver variants with >25 observations reported in COSMIC (v72, https://cancer.sanger.ac.uk/cosmic/download)[49].

**Sample processing for determination of sequence variant limit of blank**. The limit of blank was evaluated for variants of strong clinical significance in *BRAF, EGFR, KRAS, NRAS, BRCA1*, and *BRCA2* and hotspot variants of potential clinical significance. The limit of blank for all other variants of potential clinical significance was not evaluated. In total, 34 unique postmortem noncancerous samples were processed, and 29 samples were processed in duplicate, yielding a total observation count of 63 samples for the determination of the limit of the blank.

**Prediction of TMB from targeted sequencing**. The eTMB algorithm was trained using the candidate variants identified as described above in 95 FFPE-derived clinical tumor samples and 11 lung tumor cell lines. Coding variants with a high PGDx Cerebro score, >3 supporting reads, and >2% mutant allele fraction were considered as candidates for the TMB algorithm. Candidate filter sets were tested, and their performance was ranked based on the Pearson correlation of the estimated TMB score, eTMB, in mutations/Mbp, exome equivalent to the TMB measured from matched-normal WES in mutations/Mbp using a WES panel of 33.4 Mb. The set of candidate variants leading to the highest correlation to WES TMB included variants at >5% mutant allele fraction and considered both synonymous and nonsynonymous variants. Somatic variants identified with high frequency in COSMIC (v72, https://cancer.sanger.ac.uk/cosmic/download)[49] were removed to reduce bias toward common cancer driver mutations evaluated across the 505 gene panel. Common germline variants, identified through their presence at a high population frequency in dbSNP (v138)[43], ExAC (v1, https://gnomad.broadinstitute.org/downloads#exac-variants)[46], and gnomAD (v2.0.2, https://gnomad.broadinstitute.org/downloads)[47], were removed along with private germline variants identified based on variant allele frequency. Using these candidate variants, the FDA-cleared IVD (PGDx elio tissue complete) reports the TMB in mutations/Mbp using the 1.3 Mbp of coding sequence evaluated as the denominator. A log-transformed regression model was applied to this panel-based TMB (pTMB) to obtain the exome-equivalent TMB value (previously referred to as eTMB), an estimate of the TMB observed in WES. This exome-equivalent value is reported in addition to the panel-based TMB value in the Research Use Only analysis mode of the assay (PGDx elio tissue complete—RUO). The eTMB can be obtained from the pTMB with the following equation:

$$eTMB = 10^{(-0.944+1.397*\log10\,(pTMB))} \qquad (1)$$

**Retraining of TMB estimation algorithm for comparison to the ThermoFisher Oncomine assay**. The 31 FFPE NSCLC samples analyzed through elio tissue complete and the ThermoFisher Oncomine Tumor Mutation Load assay were included in the original eTMB training set. To address the bias inherent in using training samples in this head-to-head comparison, we retrained the eTMB algorithm without these 31 samples. This eTMB algorithm, rather than the eTMB algorithm described in all of the other accuracy and analytical studies, was used in the head-to-head comparison between elio tissue complete and the ThermoFisher assay.

**Processing WES through Strelka for ThermoFisher Oncomine comparison**. In order to remove the potential bias introduced in performance favoring elio tissue complete by using similar bioinformatic approaches in the WES analysis, the tumor and matched-normal WES samples were analyzed with the third-party variant caller Strelka2 (v2.9.2)[50]. The tumor and its matched normal were analyzed through the somatic workflow with the hg19 reference genome. Called variants were filtered for those designated as "PASS" in the resulting VCF file as well as having >50× coverage in the tumor, >10× coverage in the normal, <2% MAF in the normal, and >10% MAF in the tumor. High-coverage variants were annotated and filtered to remove noncoding variants.

**Identification of MSI tracts**. A reference bed file of all mononucleotide, dinucleotide, and trinucleotide tracts across hg19 was sourced and intersected with the regions of interest targeted in elio tissue complete. The coverage of each tract was evaluated across a cohort of 755 FFPE-derived clinical tumor specimens, and tracts with systematically insufficient coverage were removed from further consideration. In the cohort of 132 FFPE-derived clinical tumor specimen with known MSI status, the ability to distinguish between MSI and MSS was evaluated. Tracts that had multiple observed tract lengths across the MSS subset of the cohort were removed from further consideration due to their risk to negatively impact specificity. Tracts that had little observed difference in length between MSI and MSS cases were also removed, as they had little power to distinguish MSI status. The final 68 tracts included in the elio tissue complete panel were all mononucleotide tracts between 13 bp and 30 bp in length.

**MSI tract classification algorithm**. Reads mapping to the 68 mononucleotide tracts were assessed for indels. Standard alignment and variant calling often do not accurately identify indels in microsatellite regions due to PCR and sequencing errors in the bases following a repeat sequence. Therefore, a secondary local realignment and indel analysis was performed using seed sequences on the 5' and 3' ends of mononucleotide tracts, and reads were considered for the expanded indel analysis if: (i) the mononucleotide repeat was more than eight bases inside of the start and end of the read, (ii) the indel length was ≤12 bases from the reference length, (iii) there were no single base changes found within the repeat region, (iv) the read had a mapping score of 60, and (v) ≤20 bases of the read were soft clipped for alignment. For each mononucleotide repeat, the resulting error-corrected indel length distribution was subjected to a peak finding algorithm in which local maxima were required to be greater than the fragment counts of the adjacent lengths ± 2 bp. Identified local maxima were filtered to include only those alleles that had >5 distinct fragments at 8% or more of the absolute coverage. If the allele length was ≥2 bp shorter than the hg19 reference length, the given mononucleotide loci were classified as exhibiting instability. A total of 68 mononucleotide tracts were analyzed, and the number of tracts exhibiting instability was used as input into the ensemble MSI classifier.

**MSI mutation context scoring algorithm**. The MSI mutation context scoring algorithm used a position weight matrix model to represent the mutation signatures associated with deficient DNA mismatch repair processes, examining the context 5 base pairs upstream and downstream of a mutation and scoring at overlapping triplet positions within this 11 base-pair window[27]. To train the weight matrix, two-position probability matrices were created per each of six genomic substitutions—one representing mismatch repair-deficient tumors and one representing mismatch repair proficient tumors. The set of mismatch repair proficient tumors excluded *POLD1/POLE*-mutated samples in order to ensure the weight matrix was measuring the dMMR signature rather than solely measuring a high number of mutations. Two position probability matrices for each genomic substitution were divided elementwise, yielding six individual position weight matrices that report the log-likelihood ratio of a candidate substitution coming from an MMR-deficient tumor vs. an MMR-proficient tumor. Each mutation in each tumor was scored with the appropriate position weight matrix, and all scores were summed for each tumor, yielding an overall weight matrix score.

**Structural variant analyses**. The elio tissue complete analyses of copy number alterations and translocations were adapted from the original Digital Karyotyping (DK)[29] and Personalized Analysis of Rearranged Ends (PARE)[29] algorithms, respectively. Briefly, to identify amplifications, normalized sequence coverage in select regions of interest was compared to the sequence coverage from a set of 20 FFPE-derived noncancerous control specimens analyzed with elio tissue complete. Regions with low coverage as well as regions with insignificant differences between the sample's normalized coverage and the control coverage were filtered from analysis. Sample tumor purity was estimated in silico from somatic sequence mutations and used to derive a fold change from a diploid copy number estimate. Gene amplifications were reported when a predicted fold change above a gene-specific threshold was observed in >25% of evaluated regions for a gene. Gene amplifications were considered indeterminate if tumor purity could not be determined and the fold change passed established cutoffs for a hypothetical 20% tumor purity sample.

To identify translocations, reads mapping greater than 2 kilobases apart or mapping discordantly were identified. Low-quality reads and alignment artifacts were removed from further consideration. Alterations were annotated for gene partner and fusion status to determine if a coding strand was expected to be maintained in the resulting gene product. Reported alterations were restricted to predicted gene fusions for selected genes, and known structural variations present in the human genome or a noncancerous control database were removed. Finally, alterations with high-quality reads that passed a fusion-specific read count threshold were reported.

**Calculation of the tumor purity after dilution with normal FFPE or cell-line DNA**. Specimens from both the eTMB and MSI limit of detection studies were diluted with matched-normal DNA to achieve the desired tumor purities for evaluation. Given the variability in the quality of the tumor and matched-normal-derived DNA, the tumor and normal DNA were not consistently amplified in the same ratio, leading to observed dilution levels that did not match the intended dilution level. To account for the effects of these differences in quality, observed tumor purity was calculated through comparison of the observed sequence alteration MAFs in each dilution replicate compared to the undiluted tumor or cell-line DNA sample, taking the median ratio of the observed sequence alteration MAF in the dilution replicate to the undiluted replicate as the tumor purity, normalized to the pathological purity, if applicable.

**Statistical analyses**. Pearson and Spearman's correlations were predicted using the cor.test() function in R (v3.5.1) with the method parameter set to "Pearson" and "Spearman", respectively. The limit of blank was calculated according to the non-parametric method of determining the 95th percentile as reported in Clinical and Laboratory Standards Institute (CLSI) guideline CLSI-EP17-A section 4.1.1 equation 2. The coefficient of variation for within run, between day, between instrument, and between operator variability was calculated through a variance component analysis using SAS (v9.4). Positive and negative agreement were calculated for every possible pairwise comparison and averaged to obtain the average positive and negative percent agreement. Confidence intervals (95%) were calculated using Wilson's score interval method with a continuity correction.

**Reporting summary**. Further information on research design is available in the Nature Research Reporting Summary linked to this article.

## Data availability

The de-identified raw sequence data and associated clinical variables data generated in this study have been deposited in the European Genome-Phenome Archive (EGA) database under accession code EGAS00001005556 as indicated in Supplementary Data 3 and 9. The de-identified raw sequence data and associated clinical variables data are available under restricted access where informed consent was provided for the release and publication of raw sequence data under an Institutional Review Board approved protocol, access can be obtained by contacting the PGDx elio tissue complete Validation Data Access Committee (https://ega-archive.org/dacs/EGAC00001002278) and will be made available for a minimum of one year. The raw sequence data where informed consent was not provided for release and publication are protected and are not available due to data privacy laws. In such cases, as well as those for which raw data is made available through EGA, the processed raw sequence data can be made available through a hosted PGDx elio tissue complete user interface, access can be obtained by contacting the PGDx elio tissue complete Validation Data Access Committee (https://ega-archive.org/dacs/EGAC00001002278). The public web resources used in this paper are listed here: the Surveillance, Epidemiology, and End Results (SEER) Program (https://seer.cancer.gov); dbSNP (v138, https://www.ncbi.nlm.nih.gov/projects/SNP/snp_summary.cgi?view+summary=view+summary&build_id=138); ExAC (v1, https://gnomad.broadinstitute.org/downloads#exac-variants); gnomAD (v2.0.2, https://gnomad.broadinstitute.org/downloads); COSMIC (v72, https://cancer.sanger.ac.uk/cosmic/download); and Synapse (syn7214402, https://www.synapse.org/#!Synapse:syn7214402/wiki/405297). The remaining data are available within the Supplementary Information and Source Data files. Source data are provided with this paper.

## Code availability

Access to the previously published Cerebro[23] machine-learning framework for somatic sequence mutation discovery can be found in GitHub at https://github.com/PGDX/cerebro-paper along with the previously described MSI detection algorithm[26]. The code to generate eTMB values as well as an example TMB-H and TMB-L case can be found in GitHub at https://github.com/PGDX/tmb-paper[51].

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

## Acknowledgements

We thank members of our laboratories for critical review of this manuscript, Leila Ettehadieh and Peter Simone for assistance in data analysis, Sudhir Chowbina for contributions to the automated bioinformatics pipeline, Eric Kong for his assistance in drafting the manuscript, and Ivan Ng and Peibing Qin for technical assistance.

## Author contributions

L.A.K. wrote the manuscript, analyzed the data, and contributed to the TMB and PGDx Cerebro method development. J.R.W. wrote the manuscript, performed in silico TCGA studies, and contributed to the TMB method development. D.E.W. wrote the manuscript and developed the PGDx Cerebro and MSI substitution context scoring methods. K.M.R.G., K.C.V., D.R., C.G., E.P., and A.G. performed and supervised the analytical studies. C.M.V. analyzed the whole-exome sequencing data. J.H. led bioinformatics software development and P.M.M. led assay development. A.Z., B.M.R., J.L.P. K.D., S.J.M., and M.B.D. provided specimens along with independent validation data for the analytical studies. J.F.T., G.C.C., S.J., J.K.S., A.M., J.D., S.V.A., L.A.D., V.E.V., and M.S. wrote the manuscript, advised on experimental study design and analysis, and supervised experimental execution.

## Competing interests

J.R.W. is the founder and owner of Resphera Biosciences LLC and serves as a consultant to Personal Genome Diagnostics. K.M.R.G., K.C.V., D.R., C.G., A.G., P.M.M., J.D., S.V.A., and M.S. are employees of Personal Genome Diagnostics. J.L.P. is an employee and stockholder of PathGroup. L.D. is a member of the board of directors of Jounce Therapeutics and Epitope. He is a compensated consultant to PetDx, Innovatus CP, Se'er, Delfi, Kinnate and Neophore. He is an inventor of multiple licensed patents (to Qiagen, Exact Biosciences, and LabCorp) related to technology for circulating tumor DNA analyses and mismatch repair deficiency for diagnosis and therapy. Some of these licenses and relationships are associated with equity or royalty payments to the inventors. He holds equity in Epitope, Jounce Therapeutics, PetDx, Se'er, Delfi, Kinnate and Neophore. He divested his equity in Personal Genome Diagnostics to LabCorp in February 2022 and divested his equity in Thrive Earlier Detection to Exact Biosciences in January 2021. His spouse holds equity in Amgen. The terms of all these arrangements are being managed by Memorial Sloan Kettering in accordance with their conflict-of-interest policy. V.E.V. is a founder of Delfi Diagnostics, serves on the Board of Directors and as a consultant for this organization, and owns Delfi Diagnostics stock, which is subject to certain restrictions under university policy. Additionally, Johns Hopkins University owns equity in Delfi Diagnostics. V.E.V. divested his equity in Personal Genome Diagnostics to LabCorp in February 2022. V.E.V. is an inventor on patent applications submitted by Johns Hopkins University related to cancer genomic analyses and cell-free DNA for cancer detection that have been licensed to one or more entities, including Delfi Diagnostics, LabCorp, Qiagen, Sysmex, Agios, Genzyme, Esoterix, Ventana and ManaT Bio. Under the terms of these license agreements, the University and inventors are entitled to fees and royalty distributions. V.E.V. is an advisor to Danaher, Takeda Pharmaceuticals, and Viron Therapeutics. These arrangements have been reviewed and approved by Johns Hopkins University in accordance with its conflict-of-interest policies. The remaining authors declare no competing interests.
