## [Peer review file · Nature Communications]

Automated next-generation profiling of genomic alterations in human cancersEditorial Note: Parts of this Peer Review File have been redacted as indicated to maintain the confidentiality of unpublished data.

REVIEWER COMMENTS

Reviewer #1 (Remarks to the Author): Expert in computational cancer genomics and bioinformatics

Keefer and colleagues describe a computational approach for profiling genetic alterations, mutational burden (TMB), and microsatellite instability (MSI) from their FDA-cleared targeted cancer sequencing panel, elioTM Tissue Complete (ETC). The computational approach involves the use of machine learning algorithms, including the previously described PGDx Cerebro, and performance evaluations using data that were publicly available and generated from cell-lines and clinical FFPE samples profiled with ETC. First, the authors described the required target size of the panel for accurate determination of TMB. Next, they assessed the performance for predicting SNVs with clinical significance and SNV hotspots by comparing to existing FDA-cleared assays, MSK-IMPACT and FoundationOne, and ddPCR. For TMB analysis, the authors trained a model to predict the exome TMB (eTMB) using whole exome sequencing (WES) data as the reference; they evaluated the accuracy of the eTMB prediction model in a pan-cancer cohort of FFPE tumor samples profiled by ETC and WES. They also include a comparison to an alternative TMB assay, ThermoFisher OncoPrint Tumor Mutation Load. Finally, they describe an ensemble classifier consisting of 68 repeat-tracts and MMR signature scores to predict MSI high (H) and stable (S) status.

The manuscript is well-written and the study describes a comprehensive and mostly thorough evaluation of the performance for the series of computational pipelines to predict variants with clinical significance, TMB, and MSI in FFPE tumor samples from the same ETC panel. The authors report great performance metrics for all evaluations. However, I have several concerns which include (1) the use of existing algorithms/software for many aspects of the pipeline but new methods have no accompanying code, (2) the use of existing established platforms as the reference for validation and lack of convincing data that this approach provides improvement or an advantage compared to existing approaches, and (3) no demonstration that the multi-faceted predictions provide any improved clinical diagnostic potential. These concerns may potentially limit the novelty of this study.

1. Performance metrics for variant calling: Although the performance of the algorithms is great, the use of NPA/specificity as a metric may sometimes result in false positives being underappreciated because of the imbalance of enormous numbers of negative events. This is especially true for Supp Table 4.4, 4.5, and to some extent Supp Table 6 & 7. The positive predictive value (aka precision) will help address this, as it is also widely used for evaluating machine learning applications.

2. TMB estimation comparison: When comparing the performance between eTMB (ETC) and ThermoFisher, the Pearson correlation result of $r=0.926$ vs 0.754 may be misleading. The reference TMB is estimated from WES, which was analyzed using a similar variant calling algorithm (VariantDx) as the ETC variant calling. This may lead to bias for high performance of eTMB (ETC) and relatively penalize ThermoFisher's performance.

3. Data and code/software availability: As per the reporting standards for a research manuscript focused on computational algorithms, analysis, and performance evaluation, I was expecting to see a statement about data availability and source code, particularly for the claimed novel components and algorithms. Currently, it will be challenging to reproduce the results and conclusions of the analyses in this study based only on brief descriptions in the Methods.

Minor Comments:

1. What is the nature of the 124k in silico variants spiked in during training, aside from SNV/INDEL type? Are they somatic events with systematically selected variant allele fractions? Can you please provide more description in the Methods?

2. Extended Data Fig 5: It is not easy to tell which samples have >30% deviation in eTMB.

3. Extended Data Fig 6 and Main text for Clonality: Perhaps either the “tumor clonality” terminology is mis-used, or it is not defined properly, but the figure may not be referring to tumor clonality. The division of MAF (which I assume to be the variant allele fraction) by the tumor purity does not give the tumor clonality. Clonality is typically defined as the proportion of tumor cells harboring the variant.

4. How were the 68 repeat tracts identified? Was this from ref 26 (Georgiadis et al. 2019)? If not, please describe.

5. What algorithm was used to determine the linear decision boundary?

Reviewer #2 (Remarks to the Author): Expert in targeted sequencing assays and cancer diagnostics

“Automated machine-learning approach for next generation profiling of sequence alterations, mutation burden, and microsatellite instability in human cancers”

Kefer LA et al.

Line 28-30. The authors state: “However, the lack of standardization as well as inadequate analytical performance of laboratory and bioinformatic methods using next generation sequencing (NGS) have limited clinical adoption of these biomarkers.” Can the authors provide references to demonstrate inadequate analytical performance of laboratory and bioinformatic methods, as well limited adoption? These biomarkers are widely used and readily detectable in clinical laboratories.

Lines 63-67. Historically, yes that is correct, but somewhat misleading. NGS has been routinely used in clinical molecular laboratories for the last 5-8 years. Perhaps it would be helpful to clarify and define the difference between single analyte testing, targeted multi-analyte testing, and comprehensive genomic profiling.

Lines 72-75. References supporting the authors' statement on lack of standardization, inadequate regulatory approval, etc. would be helpful to support this statement. Much of the published literature and surveys cite factors such as limited reimbursement, questions of clinical utility, and infrastructure requirements as the major barriers to clinical adoption.

Supplementary Table 2: The total coverage across the 2370 samples ranges from 997 to 5119, the distinct coverage ranges from 242 to 2435, and the fraction of regions with >100X median distinct coverage ranges from 90-100%. Can the authors clarify what is meant by "total coverage" and "distinct coverage"? Are these average coverage and unique read coverage? Can the authors comment on the wide variability shown here? This appears to be a wider range than targeted for other comprehensive genomic profiling assays. What is the percentage of regions meeting the minimum distinct coverage per base (not the median distinct coverage) required for reliable variant calling?

Lines 130-147. It is not clear to me why the authors separate out PPA and NPA for "clinically significant SNVs", "SNVs or indels at hotspots" and all other variants. Although the "clinically significant" or hotspot alterations might be the most relevant for many targeted therapies, a high level of sensitivity and specificity is expected across the panel, as all variants are used for measuring TMB, and non-hotspot mutations may also be targetable and show potential clinical significance.

Lines 147-150 and Supplementary Table 5.

- 38 variants were not identified because the variants were present "Below ETC LOD". The ETC MAF for these variants ranges from 1.0 to 7.6%. In the Methods, lines 490-492, the authors state "Finally, reported variants must have either 4 or 6 variant observations and a MAF above 0.4% - 5.0%, depending on the level of evidence for clinical actionability and cancer driver potential." First of all, some of the missed variants were present above the 5% stated in the methods. The abstract (line 38) states high sensitivity to 3% mutant allele fraction. It is not clear from the data presented what the actual LOD is. Secondly, "clinical actionability and cancer driver potential" should not dictate what minimum LODs are. In lines 158-168 where the authors describe their LOD study, they only demonstrate LOD levels of 3-6% for some variants. It is unclear why variants as low as 0.4% are described in the Methods as being called.
- 22 variants were not detected by ETC and the resolution was "unknown". Can the authors provide any additional information? Is any resolution planned?
- 27 variants were not detected by ETC because of low variant quality. What is the reason for low variant quality? Does the assay not have sufficient coverage for these regions? Can the authors supplement their variant calling software with other software? Many of these variants are of potential clinical significance.

Line 195. "average 1,222-fold distinct coverage and 1,798-fold distinct coverage" Is this total coverage and distinct coverage?

Line 198-201. Because mutant allele fraction is one of the features considered, it would be useful to analyze whether any difference was observed in cases with lower distinct coverage. Particularly in cases with distinct coverage well below the distinct coverage in the cases used in the training set.

Lines 205-508. Would be important to examine the correlation with outlier cases (with very high TMBs) excluded. Particularly at or around the decision point for therapy (approximately 10 mutations/Mb).

Lines 259-262. These data suggest potential issues with accuracy of variant calling at 20% tumor. Suggest evaluating accuracy specifically in the subset of cases with lower tumor percentages. In some cases, this deviation in eTMB is clinically significant as it changes the patient from TMB low to TMB high. In the methods (line 542), different variant calling parameters are used for training than were established for the clinically significant mutation calls. Can the authors comment on this difference? Could this account for the lower accuracy seen for eTMB at lower tumor percentages?

Lines 406-408. Can the authors provide references supporting this statement?

General comments:

The authors mention that their assay is able to assess structural variants. However, no details relating to this in the methods, or any performance characteristics are presented.

Copy number alterations are important and clinically significant and readily identified through comprehensive genomic profiling and are reported in the two FDA cleared assays referenced in the manuscript. The authors, for example, mention ERBB2 amplification in breast cancer as an important biomarker. Can the authors comment on why they chose to omit detection and reporting of this class of important biomarkers from their assay?

Detailed Responses to Reviewer Comments

Manuscript ID: NCOMMS-21-03459-T

Automated machine-learning approach for next generation profiling of sequence alterations, mutation burden, and microsatellite instability in human cancers

Reviewer #1 (Remarks to the Author): Expert in computational cancer genomics and bioinformatics

I have several concerns which include

- (1) the use of existing algorithms/software for many aspects of the pipeline but new methods have no accompanying code
- (2) the use of existing established platforms as the reference for validation and lack of convincing data that this approach provides improvement or an advantage compared to existing approaches
- (3) no demonstration that the multi-faceted predictions provide any improved clinical diagnostic potential.

Major Comments:

1. Performance metrics for variant calling: Although the performance of the algorithms is great, the use of NPA/specificity as a metric may sometimes result in false positives being underappreciated because of the imbalance of enormous numbers of negative events. This is especially true for Supp Table 4.4, 4.5, and to some extent Supp Table 6 & 7. The positive predictive value (aka precision) will help address this, as it is also widely used for evaluating machine learning applications.

Response: We thank the reviewer for pointing out this issue. Per the reviewer's request, we have included the PPVs for all accuracy analyses in Supplementary table 4 of the revised manuscript. Supplementary Tables 6 and 7 focus on analyses of single alterations for limit of detection or healthy individuals without alterations, respectively, and would therefore not be appropriate for analyses of PPV.

2. TMB estimation comparison: When comparing the performance between eTMB (ETC) and ThermoFisher, the Pearson correlation result of $r=0.926$ vs 0.754 may be misleading. The reference TMB is estimated from WES, which was analyzed using a similar variant calling algorithm (VariantDx) as the ETC variant calling. This may lead to bias for high performance of eTMB (ETC) and relatively penalize ThermoFisher's performance.

Response: To address the reviewer's point above, we have separately re-analyzed these cases with the commonly used variant detection approach Strelka2 to obtain an independent analysis of WES sequence changes for comparison to eTMB and the ThermoFisher approach. These analyses resulted in a similarly high correlation between eTMB and the Strelka WES changes ($r=0.926$), while the ThermoFisher analyses continue to have a lower correlation ($r=0.748$). We have included these new analyses below as well as in the text (page 10) and Extended Data Fig 7 of the revised manuscript.

Extended Data Fig. 7. Comparison of TMB measurements by ETC or ThermoFisher TMB assays to WES TMB. Thirty-one NSCLC FFPE clinical tumors were processed through the ETC assay (left) and by the ThermoFisher OncoPrint Tumor Mutation Load assay (right). Using a TMB estimation algorithm that was not trained on the analyzed samples, the eTMB showed higher accuracy to WES TMB as determined by Strelka than the ThermoFisher assay (Pearson correlation = 0.926 and 0.748, respectively).

3. **Data and code/software availability:** As per the reporting standards for a research manuscript focused on computational algorithms, analysis, and performance evaluation, I was expecting to see a statement about data availability and source code, particularly for the claimed novel components and algorithms. Currently, it will be challenging to reproduce the results and conclusions of the analyses in this study based only on brief descriptions in the Methods.

Response: In the revised manuscript, we have indicated the availability of the new code for the eTMB prediction to the GitHub repository (page 23), including with example input files for TMB-high and TMB-low cases, and provided detailed methodology for development of a position weight matrix for the MSI detection approach (pages 11, 21). Separately, for existing algorithms for detection of sequence variants, MSI, copy number and rearrangement analyses, we have cited the relevant publications and included any modifications to these approaches in the Methods section of the revised manuscript (pages 21-22). Additionally, we have provided all of the candidate variants used for the eTMB determination for the eTMB validation cohort in Supplementary Table 9. Finally, we have uploaded into EGA the whole exome and targeted sequence data from all samples where such data could be shared according to IRB approvals.

Minor Comments:

1. What is the nature of the 124k in silico variants spiked in during training, aside from SNV/INDEL type? Are they somatic events with systematically selected variant allele fractions? Can you please provide more description in the Methods?

Response: For these in silico variants, we have followed a similar approach to the original PGDx Cerebro publication.¹ To address the reviewer's concern, we have included additional details on the specifics of the ETC PGDx Cerebro model in the methods section of the revised manuscript (page 17).

2. Extended Data Fig 5: It is not easy to tell which samples have >30% deviation in eTMB.

Response: In the revised manuscript, we have updated Extended Data Fig 5 to show samples with >30% deviation in eTMB as triangles rather than circles and have also identified these shape differences in the figure legend.

3. Extended Data Fig 6 and Main text for Clonality: Perhaps either the "tumor clonality" terminology is mis-used, or it is not defined properly, but the figure may not be referring to tumor clonality. The division of MAF (which I assume to be the variant allele fraction) by the tumor purity does not give the tumor clonality. Clonality is typically defined as the proportion of tumor cells harboring the variant.

Response: This figure represents an analysis of tumor heterogeneity and purity and the impact that these have on the accurate determination of TMB using our ETC approach. Specifically, we indicate on the vertical axis the tumor purity at which the TMB estimate deviated from the expected TMB by more than 30%. On the horizontal axis, we show the fraction of reads harboring a variant, corrected for tumor purity, as a rough measure of tumor clonality. Overall, we show that for those tumors where the mutations are more clonal (i.e. present in a higher fraction of tumor cells), the tumor purity which can be assessed is lower.

In the revised manuscript, we have clarified in these aspects as well as limitations to the approach (pages 9-10) and have updated the figure legends of Extended Data Figure 6.

4. How were the 68 repeat tracts identified? Was this from ref 26 (Georgiadis et al. 2019)? If not, please describe.

Response: Per the reviewer's request, in the methods of the revised manuscript, we have included a section to describe how these repeat tracts were identified ("*Identification of MSI Tracts*"). Briefly, we identified all mononucleotide, dinucleotide, and trinucleotide tracts in the regions of interest targeted in ETC. The coverage of each tract was evaluated across a cohort of 755 FFPE-derived clinical tumor specimen, and tracts with systematically insufficient coverage were removed. In the subset of cases with known MSI status, the ability to distinguish between MSI and MSS was evaluated. Tracts that had multiple observed tract lengths across the MSS subset of the cohort were removed due to their risk to specificity. Tracts which had little observed difference in length between MSI and MSS cases were also removed, as they had little distinguishing power.

5. What algorithm was used to determine the linear decision boundary?

Response: The decision boundary was developed to accurately identify MSI and MSS cases in the training data while maintaining an equal distance between the boundary and nearest points from both classes. This process is similar to a linear support vector machine classifier, but it was not fit to the training data via machine learning in order to aid in interpretability. The approach was subsequently tested in the validation study as described in the manuscript. We clarified these aspects of this approach in the revised version of the manuscript (page 11).

Reviewer #2 (Remarks to the Author): Expert in targeted sequencing assays and cancer diagnostics

Major Comments:

1. The authors mention that their assay is able to assess structural variants. However, no details relating to this in the methods, or any performance characteristics are presented. Copy number alterations are important and clinically significant and readily identified through comprehensive genomic profiling and are reported in the two FDA cleared assays referenced in the manuscript. The authors, for example, mention ERBB2 amplification in breast cancer as an important biomarker. Can the authors comment on why they chose to omit detection and reporting of this class of important biomarkers from their assay?

Response: We had initially envisioned this manuscript as focusing on SNV, TMB and MSI changes, but to address the reviewer's concern, we have now added new data and analyses describing the performance characteristics for detection of structural and copy number variants. These include analyses of 12 genes that are frequently rearranged and 16 that are frequently amplified in human cancer. We assessed the performance of these genes over >340 samples that were independently evaluated using alternative approaches. We have included these new data in the text (page 13) and new Table 1 of the revised manuscript.

Minor Comments:

1. Line 28-30. The authors state: “However, the lack of standardization as well as inadequate analytical performance of laboratory and bioinformatic methods using next generation sequencing (NGS) have limited clinical adoption of these biomarkers.” Can the authors provide references to demonstrate inadequate analytical performance of laboratory and bioinformatic methods, as well limited adoption? These biomarkers are widely used and readily detectable in clinical laboratories.

Response: We appreciate the reviewer raising this issue and agree that the text could be improved and further developed to highlight the limited adoption and use of NGS sequencing for patient care due to a variety of reasons. As the reviewer indicates here and below in item 3, these reasons include limited reimbursement and infrastructure requirements, long turn-around time of send-out NGS tests to be useful for patient care, and institutional preference for local control and aggregation of sensitive patient data. Additionally, the lack of availability of FDA approved tests (prior to PGDx’s ETC) would require individual CLIA labs to spend significant resources to develop IDT tests on their own to be utilized locally, an approach that has been challenging for many labs to complete. As FDA approval is linked to Medicare reimbursement, the availability of an FDA approved comprehensive tumor profile test now permits reimbursement at a level that allows local labs to perform these tests in their facilities. We have included multiple references that indicate limited adoption of NGS testing (~20% of metastatic patients)²⁻³, preference for institutions to house data locally⁴, and the connection between FDA approval and Medicare reimbursement⁵ in the revised manuscript. We have also reworded the abstract and introduction of the revised manuscript to clarify these points.

2. Lines 63-67. Historically, yes that is correct, but somewhat misleading. NGS has been routinely used in clinical molecular laboratories for the last 5-8 years. Perhaps it would be helpful to clarify and define the difference between single analyte testing, targeted multi-analyte testing, and comprehensive genomic profiling.

Response: We thank the reviewer for raising this point. We have clarified single analyte testing and comprehensive genomic profiling in the revised version of the manuscript to reference the historical development of these tests in molecular laboratories (page 3). We have not discussed multi-analyte testing, as this topic is outside the scope of this manuscript.

3. Lines 72-75. References supporting the authors’ statement on lack of standardization, inadequate regulatory approval, etc. would be helpful to support this statement. Much of the published literature and surveys cite factors such as limited reimbursement, questions of clinical utility, and infrastructure requirements as the major barriers to clinical adoption.

Response: We appreciate the reviewer raising this issue and agree that the text could be improved and further developed to highlight the limited adoption and use of NGS sequencing for patient care due to a variety of reasons. As the reviewer indicates here and above in item 1, these reasons include limited reimbursement and infrastructure requirements, as well as long turn-around time of send-out NGS tests to be useful for patient care, as well as institutional preference of local control and aggregation of sensitive patient data. Additionally, the lack of

availability of FDA approved tests (prior to PGDx's ETC) would require individual CLIA labs to spend significant resources to develop LDT tests on their own to be utilized locally, an approach that has been challenging for many labs to complete. As FDA approval is linked to Medicare reimbursement, the availability of an FDA approved comprehensive tumor profile test now permits reimbursement at a level that allows local labs to perform these tests in their facilities. We have included multiple references that indicate limited adoption of NGS testing (~20% of metastatic patients)²⁻³, preference for institutions to house data locally⁴, and the connection between FDA approval and Medicare reimbursement⁵ in the revised manuscript. We have also reworded the abstract and introduction of the revised manuscript to clarify these points.

4. Supplementary Table 2: The total coverage across the 2370 samples ranges from 997 to 5119, the distinct coverage ranges from 242 to 2435, and the fraction of regions with >100X median distinct coverage ranges from 90-100%.
 - a. Can the authors clarify what is meant by "total coverage" and "distinct coverage"? Are these average coverage and unique read coverage?

Response: We defined total coverage as the average total number of reads sequenced at any position across the regions of interest in the elio tissue complete panel. The distinct coverage is the average number of unique DNA molecules sequenced at any position across the targeted regions of interest. A substantial difference between the total coverage and unique coverage can indicate a limited amount of amplifiable DNA in an analyzed sample. We have clarified these items in the footnote of Supplementary Table 2 and in the Methods of the revised manuscript (page 17).

- b. Can the authors comment on the wide variability shown here? This appears to be a wider range than targeted for other comprehensive genomic profiling assays.

Response: The wide variability in coverage is a reflection of the wide variability in sample quality of the samples analyzed in the study – these may be related to formalin fixation, specimen age, and sample size (e.g. biopsies versus surgical resections). When one limits these analyses to the same input material (e.g. replicate samples in the analytical validation), the variability is much lower, indicating high reproducibility for the ETC test. Overall, these results indicate that despite sample variables that may occur in real-world laboratory settings, the ETC assay achieves sufficient total and distinct coverage to achieve high performance for detection of genomic alterations assessed. We have clarified these issues in the revised text of the manuscript (pages 10 and 17).

- c. What is the percentage of regions meeting the minimum distinct coverage per base (not the median distinct coverage) required for reliable variant calling?

Response: The ETC assay is a hybrid capture based system, employing 120bp probes for targeted in-solution capture of the regions of interest. This design results in coverage that tends to increase towards the middle of the targeted region based on the probe tiling coordinates. Given this, we set a coverage QC threshold of $\geq 90\%$ of regions with a median distinct coverage of $\geq 100x$, as it would be unlikely that with this type of capture approach individual bases would not be sequenced (this is also in

alignment with the representative variant approach taken for system validation, as recommended by the FDA CDRH⁶). Finally, while the median distinct coverage is required to be at least 100x for a given region to be considered, by definition, at least 50% of the bases in any region will have $\geq 100x$ distinct coverage.

5. Lines 130-147. It is not clear to me why the authors separate out PPA and NPA for “clinically significant SNVs”, “SNVs or indels at hotspots” and all other variants. Although the “clinically significant” or hotspot alterations might be the most relevant for many targeted therapies, a high level of sensitivity and specificity is expected across the panel, as all variants are used for measuring TMB, and non-hotspot mutations may also be targetable and show potential clinical significance.

Response: Due to the intended use of the ETC assay for pan-solid tumor profiling together with the current FDA guidance⁶ regarding how these different classes of variants should be reported, we felt it important to differentiate performance across each category of alteration given the implications for a false positive or negative result in each setting. By design, we expect higher sensitivity and specificity for variants for which targeted therapies may be administered, and we thought it meaningful to highlight the high accuracy for these specific, clinically relevant variants. In contrast, variants with unknown significance, while important individually for the TMB score calculation, are less relevant for a patient’s treatment course (it is also important to note that clinically relevant and hotspot variants are removed from the TMB calculation as these are less useful for providing a measure of overall mutation frequencies). Overall, we have shown that with the observed level of sensitivity and specificity achieved across all categories of sequence variants, we were able to determine the eTMB with high accuracy compared to WES.

In response to this comment, we have added a description of the variants in each of these categories with the reference to the FDA guidance above in the revised version of the methods section of the manuscript (page 19).

6. Lines 147-150 and Supplementary Table 5.
 - a. 38 variants were not identified because the variants were present “Below ETC LOD”. The ETC MAF for these variants ranges from 1.0 to 7.6%. In the Methods, lines 490-492, the authors state “Finally, reported variants must have either 4 or 6 variant observations and a MAF above 0.4% - 5.0%, depending on the level of evidence for clinical actionability and cancer driver potential.”
 - i. First of all, some of the missed variants were present above the 5% stated in the methods. The abstract (line 38) states high sensitivity to 3% mutant allele fraction. It is not clear from the data presented what the actual LOD is.

Response: The LOD for a given variant is position-specific and dependent on the category of alteration as outlined in the methods section of the manuscript (4 or 6 variant observations and a MAF above 0.4% - 5.0%, depending on the level of evidence for clinical actionability and cancer driver potential). Additionally, the LOD (95%) will always be higher than the threshold utilized for variant calling and can be influenced by several factors,

including variability in distinct coverage and underlying genomic complexity. For these specific examples highlighted by the reviewer in the supplementary table, either the MAF or the number of mutant reads fell below the threshold, depending on the variant classification.

To clarify these points, we have revised the language in Supplementary Table 5, modifying “Below ETC LOD” to “Below ETC Thresholds” to better represent the explanation as to why these variants were filtered. Additionally, we have clarified this in the abstract to further elaborate on this point.

- ii. Secondly, “clinical actionability and cancer driver potential” should not dictate what minimum LODs are. In lines 158-168 where the authors describe their LOD study, they only demonstrate LOD levels of 3-6% for some variants. It is unclear why variants as low as 0.4% are described in the Methods as being called.

Response: Due to the intended use of the ETC assay for pan-solid tumor profiling, together with the current FDA guidance⁶ regarding how these different classes of variants should be reported, we felt it important to maximize sensitivity and specificity for variants with clinical actionability and cancer driver potential through selection of alterations known to be mutated in cancer. These thresholds were established and validated based on the limit of blank study as reported for a given genomic position (for variants with clinical actionability) and across genomic positions (for variants with unknown clinical actionability). These analyses have resulted in an improved LOD for these classes of alterations while maintaining specificity due to the more limited region assessed and the higher *a priori* likelihood that these were *bona fide* somatic mutations (the observation of an uncharacterized somatic variant requires further evidence to maintain performance). This can be particularly important for accurate somatic mutation analyses when the tumor content is low and the impact of a false positive or negative alteration for variants with clinical actionability is higher.

- b. 22 variants were not detected by ETC and the resolution was “unknown”. Can the authors provide any additional information? Is any resolution planned?

Response: While every attempt was made to resolve the 22 variants not detected in the sequencing data by ETC, we were unable to successfully annotate each variant due to the more limited data received from the orthogonal tests. For example, differences in variant annotation, reporting (e.g. MAF), level of evidence, and thresholds for variant calling have resulted in our inability to definitively resolve each discrepancy. However, to minimize the clinical impact of this topic for the reader, we were able to resolve all discrepancies for variants that were clinically actionable.

To clarify this point in the revised manuscript, we have updated the resolution status of these 22 variants to “Insufficient data to resolve” to better reflect their appropriate status.

- c. 27 variants were not detected by ETC because of low variant quality. What is the reason for low variant quality? Does the assay not have sufficient coverage for these regions? Can the authors supplement their variant calling software with other software? Many of these variants are of potential clinical significance.

Response: We thank the reviewer for providing this comment. Low variant quality does not refer to the variant position or assay, but that the variant that did not meet the confidence threshold from the Cerebro machine learning variant detection algorithm. Cerebro evaluates 60+ features when determining the confidence score, and as such, a low Cerebro confidence score can be derived from a combination of low performing features without assignment to any one unique feature.

We have listed variants as potentially clinically actionable according to FDA guidelines⁶ and have clarified the language in Supplementary Table 5.

7. Line 195. “average 1,222-fold distinct coverage and 1,798-fold distinct coverage” Is this total coverage and distinct coverage?

Response: Both values are distinct coverage. The two values correspond to the distinct coverage of the clinical tumor samples (1,222-fold) and the cell line samples (1,798-fold).

8. Line 198-201. Because mutant allele fraction is one of the features considered, it would be useful to analyze whether any difference was observed in cases with lower distinct coverage. Particularly in cases with distinct coverage well below the distinct coverage in the cases used in the training set.

Response: The reviewer raises an excellent point in that we may expect to underestimate TMB for samples with reduced distinct coverage. However, given the high level of distinct coverage even in our lower coverage cases, there was no significant correlation observed between the absolute error for TMB estimation (eTMB - WES TMB) and the average distinct coverage for any given case, as seen below.

9. Lines 205-508. Would be important to examine the correlation with outlier cases (with very high TMBs) excluded. Particularly at or around the decision point for therapy (approximately 10 mutations/Mb).

Response: We thank the reviewer for this feedback, and as such, have included a Spearman correlation analysis in the revised manuscript, which represents a rank-based approach to minimize the impact of outlier samples compared to the Pearson methodology (Spearman rho = 0.870, $p < 0.0001$; Figure 3c).

10. Lines 259-262.

- a. These data suggest potential issues with accuracy of variant calling at 20% tumor. Suggest evaluating accuracy specifically in the subset of cases with lower tumor percentages. In some cases, this deviation in eTMB is clinically significant as it changes the patient from TMB low to TMB high.

Response: We appreciate the reviewer's concern, and to address this point have performed a supplementary analysis comparing the absolute error of the ETC eTMB estimate to WES-derived TMB across the range of tumor content assessed in this pan-tumor study. As demonstrated below, across the cohort of >300 samples from the pan-cancer validation study, the absolute error from the reference WES value was independent of tumor content.

- b. In the methods (line 542), different variant calling parameters are used for training than were established for the clinically significant mutation calls. Can the authors comment on this difference? Could this account for the lower accuracy seen for eTMB at lower tumor percentages?

Response: While different variant calling parameters were used to identify candidate variants for the eTMB algorithm compared to clinically actionable sequence variants, the thresholds used for eTMB candidate variant identification did not change between the training and validation studies for the eTMB method. We believe the lower accuracy for eTMB at lower tumor purities is due to the presence of candidate variants at or near the limit of detection used for TMB identification (5%), performance of which was verified in the LOD study for eTMB and clinically actionable sequence variants.

11. Lines 406-408. Can the authors provide references supporting this statement?

“Additionally, differences in sequencing instruments and standardization for targeted panels can give rise to variation in TMB determination, making objective assessment problematic.”

Response: We have added a reference from the Friends of Cancer Research TMB Harmonization Project⁷ to support this statement.

References:

1. Wood, D. E. et. al. A machine learning approach for somatic mutation discovery. *Sci. Transl. Med.* **10**, eaar7939 (2018).
2. Freedman, A. N. et. al. Use of next-generation sequencing tests to guide cancer treatment: Results from a nationally representative survey of oncologists in the United States. *JCO Precis. Oncol.* (2018).
3. Vadas, A. Immuno-oncology is making pharma step up its diagnostics game. *In Vivo* **36**, 2-8 (2018).
4. Messner, D. A. et al. Barriers to clinical adoption of next generation sequencing: Perspectives of a policy Delphi panel. *Appl. Transl. Genom.* **10**, 19-24 (2016).
5. Centers for Medicare & Medicaid Services. National Coverage Determination (NCD) for Next Generation Sequencing (NGS) (90.2). Version 2. Effective 1/27/2020.
6. United States Food and Drug Administration. CDRH's approach to tumor profiling next generation sequencing tests. <https://www.fda.gov/media/109050/download>. Last accessed August 4, 2021.
7. Merino, D. M. et. al. Abstract 5671: Alignment of TMB measured on clinical samples: Phase IIB of the Friends of Cancer Research TMB Harmonization Project. *Cancer Res.* **80**, 5671 (2020).

REVIEWER COMMENTS

Reviewer #1 (Remarks to the Author):

1. Since this manuscript is largely related to a reporting of an “automated machine learning” approach for analyzing tumor sequencing data, the authors should provide the machine learning code as described in the manuscript. In the revised manuscript, eTMB equation code is provided in a short R script but many aspects are missing, including the pipeline (Figure 1), MSI scoring algorithm, and the machine learning components. Especially since most of the pipeline is based on already published methods, then including code/software for all of the new methodology is essential. Inclusion of the overall pipeline would be ideal as well, which will maximize the usefulness and reproducibility of this work for the research community.

2. These previous general concerns have not been addressed:

a. The use of existing established platforms as the reference for validation and lack of convincing data that this approach provides improvement or an advantage compared to existing approaches.

b. No demonstration that the multi-faceted predictions provide any improved clinical diagnostic potential.

Reviewer #2 (Remarks to the Author):

Thank you for the detailed response. The authors have done an excellent job addressing the comments raised.

Automated machine-learning approach for next generation profiling of sequence and structural alterations, mutation burden, and microsatellite instability in human cancers

Keefer et al.,

Response to Reviewer Comments

- Reviewer #1 (Remarks to the Author)
 - Since this manuscript is largely related to a reporting of an “automated machine learning” approach for analyzing tumor sequencing data, the authors should provide the machine learning code as described in the manuscript. In the revised manuscript, eTMB equation code is provided in a short R script but many aspects are missing, including the pipeline (Figure 1), MSI scoring algorithm, and the machine learning components. Especially since most of the pipeline is based on already published methods, then including code/software for all of the new methodology is essential. Inclusion of the overall pipeline would be ideal as well, which will maximize the usefulness and reproducibility of this work for the research community.

Response: We appreciate the Reviewer’s concern that access to the algorithms and complete bioinformatics pipeline results were not provided in the previously submitted manuscript, which limits the utility of this study for the broader community. Therefore, we have provided a direct link to our previously published GitHub repository in the revised “Code and Data Availability” section of the Methods (Wood DE, White JR, Georgiadis A, et al. A machine learning approach for somatic mutation discovery. *Sci Transl Med.* 2018;10(457):eaa7939. doi:10.1126/scitranslmed.aar7939; <http://github.com/PGDX/cerebro-paper>) where the machine learning framework for somatic sequence mutation discovery was described in detail, along with the relevant reference to the MSI detection algorithm (Georgiadis A, Durham JN, Keefer IA, et al. Noninvasive Detection of Microsatellite Instability and High Tumor Mutation Burden in Cancer Patients Treated with PD-1 Blockade. *Clin Cancer Res.* 2019;25(23):7024-7034. doi:10.1158/1078-0432.CCR-19-1372). Additionally, we have made our automated bioinformatics pipeline results available for representative datasets where informed consent was provided under IRB protocol, which follows a similar mechanism for access to the raw data we have deposited with EGA (REDACTED).

We feel that these significant additions to a further revised manuscript, both in terms of reference to prior publications and source code of specific algorithms as well as through enabling access to the automated bioinformatics pipeline results, address the Reviewer’s principal concerns. Furthermore, these revisions exceed the transparency provided by prior publications of commercial diagnostic laboratory developed tests, where bioinformatics pipeline access and source code were not provided:

- i. Frampton GM, Fichtenholtz A, Otto GA, et al. Development and validation of a clinical cancer genomic profiling test based on massively parallel DNA sequencing. *Nat Biotechnol.* 2013;31(11):1023-1031. doi:10.1038/nbt.2696
- ii. Clark TA, Chung JH, Kennedy M, et al. Analytical Validation of a Hybrid Capture-Based Next-Generation Sequencing Clinical Assay for Genomic Profiling of Cell-Free Circulating Tumor DNA. *J Mol Diagn.* 2018;20(5):686-702. doi:10.1016/j.jmoldx.2018.05.004
- iii. Ianman RB, Mortimer SA, Zill OA, et al. Analytical and Clinical Validation of a Digital Sequencing Panel for Quantitative, Highly Accurate Evaluation of Cell-Free Circulating Tumor

**DNA. PLoS One. 2015;10(10):e0140712. Published 2015 Oct 16.
doi:10.1371/journal.pone.0140712**

- These previous general concerns have not been addressed:
 - a. The use of existing established platforms as the reference for validation and lack of convincing data that this approach provides improvement or an advantage compared to existing approaches.
 - b. No demonstration that the multi-faceted predictions provide any improved clinical diagnostic potential.

Response: We have performed a systematic review of existing FDA cleared multigene solid tumor profiling diagnostic assays to summarize the improvements afforded through elio tissue complete for comprehensive genomic profiling of a broad range of alteration types, including sequence and structural alterations, as well as novel genomic signatures, such as TMB and MSI. Additionally, based on these additional biomarker features assessed, we have summarized the clinical diagnostic utility across FDA-recognized biomarkers predictive of response to FDA-approved drugs, standard of care biomarkers recommended by professional guidelines predictive of response to FDA-approved drugs, and standard of care biomarkers predictive of resistance to FDA-approved drugs for each relevant gene and tumor type. These data are now summarized in a new Supplementary Table 2 in the revised manuscript.

REVIEWERS' COMMENTS

Reviewer #1 (Remarks to the Author):

The authors have addressed all of my concerns. Thank you for your efforts to provide the necessary details of the various sources of code for methods described in this manuscript.

Automated next-generation profiling of genomic alterations in human cancers

Keefer et al.,

Response to Reviewer Comments

- **Reviewer #1 (Remarks to the Author):**

The authors have addressed all of my concerns. Thank you for your efforts to provide the necessary details of the various sources of code for methods described in this manuscript.

Response: We thank the reviewer for their critical review of our manuscript, which we feel has strengthened the scientific foundation of this study and will enable broader impact to the oncology and diagnostics community.